# A somatic proteoglycan controls Notch-directed germ cell fate

Sandeep Gopal [1✉], Aqilah Amran [1], Andre Elton [1], Leelee Ng[1] & Roger Pocock [1✉]

Communication between the soma and germline optimizes germ cell fate programs. Notch receptors are key determinants of germ cell fate but how somatic signals direct Notch-dependent germ cell behavior is undefined. Here we demonstrate that SDN-1 (syndecan-1), a somatic transmembrane proteoglycan, controls expression of the GLP-1 (germline proliferation-1) Notch receptor in the *Caenorhabditis elegans* germline. We find that SDN-1 control of a somatic TRP calcium channel governs calcium-dependent binding of an AP-2 transcription factor (APTF-2) to the *glp-1* promoter. Hence, SDN-1 signaling promotes GLP-1 expression and mitotic germ cell fate. Together, these data reveal SDN-1 as a putative communication nexus between the germline and its somatic environment to control germ cell fate decisions.

[1] Department of Anatomy and Developmental Biology, Monash Biomedicine Discovery Institute, Monash University, Melbourne, Victoria 3800, Australia.
✉email: sandeep.gopal@monash.edu; roger.pocock@monash.edu

Syndecans are a family of transmembrane proteoglycans that play important roles in metazoan development, regeneration, tissue repair, and homeostasis[1]. Syndecans consist of conserved cytoplasmic and transmembrane domains and more divergent ectodomains that carry differentially sulfated glycosaminoglycan (GAG) chains[2]. These GAG chains consist of heparan sulfate (HS) and chondroitin sulfate (CS) that can interact with ligands, extracellular matrix proteins, cytokines, growth factors, and small molecules to coordinate signaling, cell adhesion, and host defense mechanisms[2]. Previously, we showed that syndecans control intrinsic cell behavior by regulating transient receptor potential canonical (TRPC) $Ca^{2+}$ channels to maintain optimal $Ca^{2+}$ influx over the plasma membrane[3]. Mechanistically, syndecans induce protein kinase C-alpha (PKCα)-dependent phosphorylation of a conserved serine residue in the cytoplasmic domain of TRPC channels[3]. This phosphorylation event closes the TRPC channel and thereby limits $Ca^{2+}$ influx[3]. As $Ca^{2+}$ can act as a second messenger, the control of $Ca^{2+}$ influx by syndecans may enable communication between the environment, extracellular matrix, and cytoplasmic/nuclear factors.

*Caenorhabditis elegans* encodes a single syndecan (SDN-1) that controls neuronal development, cell migration, animal behavior, and spindle orientation[4–6]. Neuronal defects caused by loss of SDN-1 are suppressed by the removal of TRPC channels, revealing that $Ca^{2+}$ regulation is an evolutionarily conserved function for syndecans[3]. Early studies of a *C. elegans sdn-1* loss-of-function mutant revealed a reduction in brood size, suggesting a role for SDN-1 in progeny generation and germline function[4]. The adult *C. elegans* hermaphrodite generates ~300 offspring from two syncytial germline tubes, within which germ cell nuclei are partially enclosed by membranes[7]. The germline is organized in a distal-to-proximal manner, with mitotically-dividing cells (including self-renewing germline stem cells—GSCs) within the distal progenitor zone (PZ), and proximal differentiated gametes[7]. As cells move through the germline from distal to proximal, they are influenced by distinct genetic programs that control their fate.

A critical regulator of germ cell fate in *C. elegans* is the GLP-1 Notch receptor, which is required for establishing and maintaining the stem cell pool within the PZ[7–9]. In this context, a single-celled mesenchymal stem cell niche, called the distal tip cell (DTC), expresses the Notch ligands LAG-2 and APX-1 that trigger GLP-1 cleavage[10–12]. The GLP-1 Notch intracellular domain that is released by this cleavage transcriptionally regulates downstream targets, including the redundantly-acting genes *lst-1* (lateral signaling target) and *sygl-1* (synthetic Glp), to maintain a pool of GSCs within the PZ[13]. Accurate regulation of GLP-1 expression and processing is critical for germline development: reduced GLP-1 expression causes germ cells to exit the mitotic cell cycle and inappropriately enter meiosis, whereas elevated GLP-1 signaling results in a germline tumor that is replete with mitotic cells[8,14]. Although much attention has been focused on deciphering mechanisms regulating Notch-ligand interactions and Notch processing, how Notch receptor transcription is controlled is not understood.

Here, we report a critical function for SDN-1 in regulating germ cell fate by controlling GLP-1 expression. We find that SDN-1 acts non-cell autonomously from somatic gonadal sheath cells to promote GLP-1-dependent germ cell fate. Further, we reveal an important role for $Ca^{2+}$ regulation in germ cell development as removal of the TRPC $Ca^{2+}$ channel, TRP-2, suppresses germ cell fate defects caused by loss of SDN-1. Genetically downstream of SDN-1, we find that *glp-1* transcription is controlled through a deeply conserved promoter motif directly bound and regulated by the APTF-2 transcription factor. The binding of APTF-2 to the *glp-1* promoter is abrogated by elevated $Ca^{2+}$, thus

providing a mechanistic link between the SDN-1/TRP-2 axis and GLP-1 transcriptional regulation. Together, our data reveal a mechanism for soma-germline communication to control germ cell fate, where SDN-1 acts from somatic cells to control GLP-1 expression and germ cell behavior. This mode of regulation may enable interpretation and communication of extra-germline environmental cues, and thus optimize germ cell proliferation and progeny generation in ephemeral habitats and in pathophysiological conditions.

## Results

**SDN-1 controls germ cell proliferation.** In *C. elegans*, SDN-1 loss-of-function results in reduced progeny production, suggesting aberrant germline function (Fig. S1A)[4]. To examine the in vivo importance of SDN-1 in the germline, we quantified germ cell number using three-dimensional germline reconstruction[15]. Using two independently-derived *sdn-1* deletion alleles, we found that loss of SDN-1 reduces the number of mitotically-dividing germ cells within the PZ, when compared to age-synchronized wild-type animals (Fig. 1a, b). Furthermore, PZ length was shorter in *sdn-1* mutants (~16–18 cell diameters) compared to wild-type animals (~20 cell diameters) (Figs. 1a and S1B). As SDN-1 functions during development to control multiple cell migration events, including DTC migration[4,16], we investigated whether post-developmental knockdown of *sdn-1* also causes a reduction in PZ germ cell number. We found that RNAi-mediated interference (RNAi) knockdown of *sdn-1* for 16 h from the L4 larval stage caused a reduction in PZ cell number but no detectable DTC migration defect (Fig. S1C–E). These data support a post-developmental function for SDN-1 in controlling germ cell fate. To assess the proliferative state of PZ cells in *sdn-1(zh20)* animals, we detected cells in the M phase by staining germlines with an antibody against phospho-histone H3 (Fig. 1c). We found that the number of M phase cells in *sdn-1(zh20)* animals were reduced by ~50% compared to wild-type (Fig. 1c, d). These data reveal an important function for SDN-1 in regulating distal germ cell behavior.

Expression of *sdn-1* in the germline or associated somatic tissues has not previously been observed with transgenic reporters[4,5,17]. However, spatiotemporal sequencing analysis detected low-level *sdn-1* mRNA in hermaphrodite gonadal tissue[18,19]. To examine *sdn-1* expression, we generated a *sdn-1p::gfp* reporter line using a 5061 bp promoter (Fig. 1e, f). We found that this *sdn-1* promoter drove GFP expression in both the DTC and gonadal sheath (Fig. 1e, f). To characterize where SDN-1 acts to control germ cell behavior, we used defined tissue-specific promoters to express a single copy of *sdn-1* cDNA, using miniMos-mediated genomic insertion (Fig. 1g, h)[20]. We expressed *sdn-1* cDNA in either the germline (*mex-5* promoter), DTC (*lag-2* promoter), or gonadal sheath (*lim-7* promoter) of *sdn-1(zh20)* mutant animals and quantified PZ cell number (Fig. 1g, h)[12,21,22]. We obtained two independent integrated lines for each single-copy transgene and found that only expressing *sdn-1* in the gonadal sheath rescued the *sdn-1(zh20)* PZ germ cell number phenotype (Fig. 1h). Similarly, the expression of *sdn-1* in the gonadal sheath rescued the number of M-phase cells in *sdn-1(zh20)* animals to wild-type levels (Fig. S2A). Examination of the gonadal sheath structure with the *lim-7p::GFP* transgene revealed no overt defect in *sdn-1(zh20)* animals (Fig. 1i). Together, these data show that SDN-1 acts non-cell autonomously from somatic gonadal sheath cells to control germ cell proliferation.

The gonadal sheath consists of somatic cells that cover the surface of the germline meiotic region that also send processes into the proximal part of the PZ[22–24]. Syndecans have established roles in regulating metazoan cell behavior through binding to

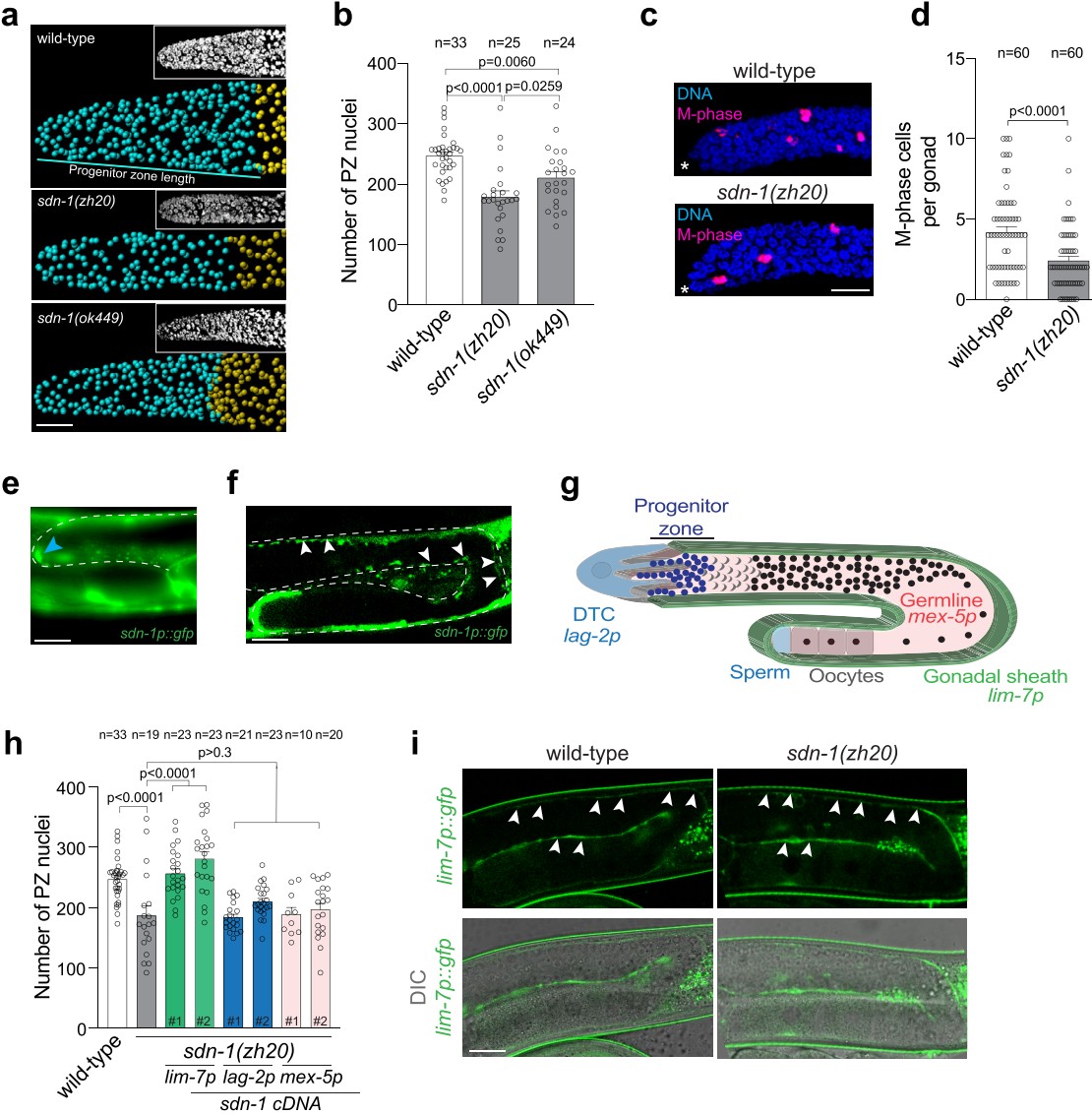

**Fig. 1 SDN-1 controls germ cell proliferation from the somatic sheath. a**, **b** 3D germline analysis images (**a**) and quantification (**b**) of germline progenitor zone (PZ) nuclei (cyan spheres) in wild-type, *sdn-1(zh20)*, and *sdn-1(ok449)* adult hermaphrodites. Insets in (**a**) = Confocal image of DAPI staining. 3D models are generated in Imaris. Transition zone nuclei are represented as gold spheres. Data were expressed as mean ± s.e.m. and statistical significance was assessed by ordinary one-way ANOVA. Scale bar = 10 µm. **c**, **d** Immunostaining images (**c**) and quantification (**d**) of M-phase cells in distal germlines in wild-type and *sdn-1(zh20)* adult hermaphrodites. Germlines stained with DAPI to visualize DNA (blue) and anti-phospho-histone H3 (pH3) to visualize M-phase chromosomes (magenta). Data were expressed as mean ± s.e.m. and statistical significance was assessed by two-tailed Welch's *t*-test. Scale bar = 10 µm. Asterisks, the distal end of the gonad. **e**, **f** Expression pattern of the *sdn-1p::gfp* transgene in the DTC (**e**-blue arrowhead) and gonadal sheath (**f**-white arrowheads). Dashed line = germline. Scale bars 20 µm. **g** Schematic of one *C. elegans* germline arm showing expression domains of promoters used in *sdn-1(zh20)* rescue experiments. **h** Quantification of PZ nuclei in wild-type, *sdn-1(zh20)* adult hermaphrodites ± single-copy transgenic expression of *sdn-1* cDNA under tissue-specific promoters (*lag-2p* - DTC, *mex-5p* - germline, *lim-7p* gonadal sheath). Independent transgenic expression lines for each tissue-specific transgene are represented by a #. Data were expressed as mean ± s.e.m. and statistical significance was assessed by ordinary one-way ANOVA. **i** Fluorescence images of *lim-7p::gfp* in the gonadal sheath (white arrowheads) in wild-type (left images) and *sdn-1(zh20)* (right images) adult hermaphrodites. Upper images (GFP) and lower images (DIC and GFP). Twenty germlines were examined in duplicate. Scale bar = 20 µm. For all panels, *n* refers to the number of animals analyzed. *P* values as indicated. Source data are provided as a Source Data file.

extracellular ligands via GAG chains that are attached to three conserved serine residues in their ectodomain[25,26]. We, therefore, investigated whether GAG chain attachment is important for SDN-1 regulation of PZ germ cell number. We mutated all three SDN-1 GAG attachment sites from serine to alanine (SDN-1-AAA) and expressed a single copy of SDN-1-AAA in the gonadal sheath of *sdn-1(zh20)* animals (Fig. S2B, D)[17]. We found that, unlike wild-type SDN-1, GAG chain-deficient SDN-1-AAA was unable to rescue the PZ germ cell length or M-phase cell phenotype of *sdn-1(zh20)* animals (Fig. S2C, D). Collectively,

these data show that SDN-1 acts from the gonadal sheath to control PZ germ cell behavior and requires the attachment of extracellular GAG chains to perform this function.

**SDN-1 controls germ cell fate through GLP-1/Notch.** The GLP-1 Notch receptor is a major regulator of germ cell fate in the *C. elegans* germline[7]. GLP-1 depletion causes germ cells to prematurely exit the mitotic cell cycle and as such reduces PZ cell number[7]. We asked whether *sdn-1* acts in the same genetic

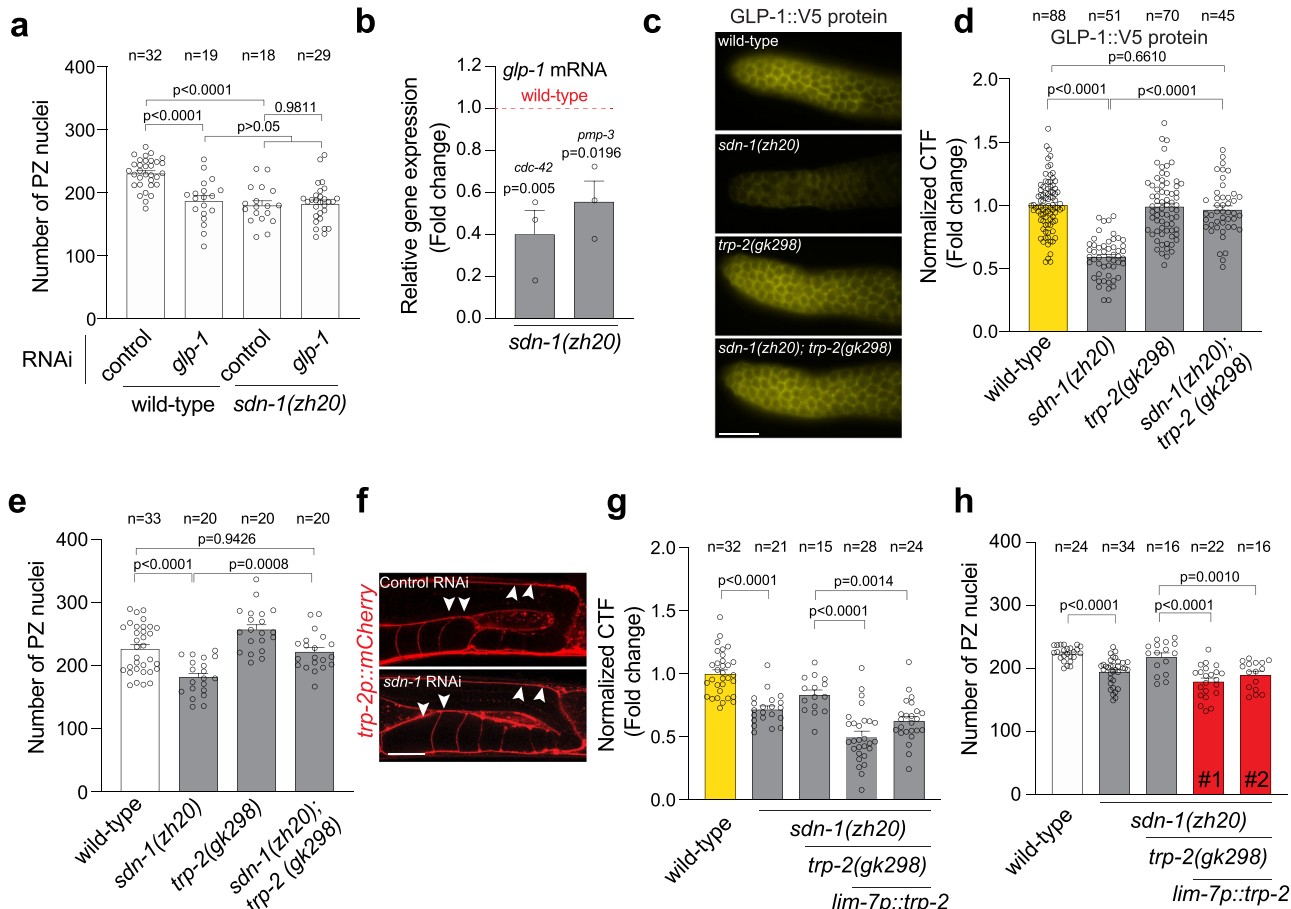

**Fig. 2 SDN-1 promotes GLP-1/Notch expression to control mitotic germ cell fate. a** Quantification of PZ nuclei in wild-type and *sdn-1(zh20)* adult hermaphrodites grown on control (L4440 vector) and *glp-1* RNAi bacteria. Data were expressed as mean ± s.e.m. and statistical significance was assessed by ordinary one-way ANOVA. **b** Quantification of *glp-1* mRNA by qPCR in *sdn-1(zh20)* animals compared to wild-type. Two independent reference genes (*cdc-42*) and (*pmp-3*) were used to normalize expression. RNA was isolated from dissected germlines. Data were expressed as mean ± s.e.m. and statistical significance was assessed by ordinary one-way ANOVA. independent biological triplicate RNA samples for each genotype. **c, d** Immunofluorescence images (**c**) and quantification (**d**) of GLP-1::V5 expression in the PZ of wild-type, *sdn-1(zh20)*, *trp-2(gk298)*, and *sdn-1(zh20); trp-2(gk298)* adult hermaphrodites. Data were expressed as mean ± s.e.m. and statistical significance was assessed by ordinary one-way ANOVA. Scale bar = 10 μm. **e** Quantification of PZ nuclei in wild-type, *sdn-1(zh20)*, *trp-2(gk298)*, and *sdn-1(zh20); trp-2(gk298)* adult hermaphrodites. Data were expressed as mean ± s.e.m. and statistical significance was assessed by ordinary one-way ANOVA. **f** Fluorescence images of *trp-2p::mCherry* in the gonadal sheath (white arrowheads) in control (upper image) and *sdn-1* RNAi (lower image) adult hermaphrodites. Twenty germlines were examined in duplicate. Scale bar 20 μm. **g, h** Quantification of GLP-1::V5 expression in the PZ (**g**) and PZ nuclei number (**h**) of wild-type, *sdn-1(zh20)*, and *sdn-1(zh20); trp-2(gk298)* (± *lim-7p::trp-2 cDNA*) adult hermaphrodites. Data were expressed as mean ± s.e.m. and statistical significance was assessed by ordinary one-way ANOVA. For all panels, *n* refers to the number of animals analyzed. *P* values as indicated. Source data are provided as a Source Data file.

pathway as *glp-1* to control PZ cell number. We achieved partial knockdown of *glp-1* for 16 h from the L4 stage using RNAi and analyzed young adult hermaphrodites (Fig. S3). As shown previously, decreased *glp-1* expression in wild-type animals reduces PZ cell number (Fig. 2a)[8]. However, *glp-1* RNAi treatment did not further reduce PZ cell number of *sdn-1(zh20)* mutant animals suggesting that *sdn-1* and *glp-1* act in the same genetic pathway (Fig. 2a).

To investigate the potential mechanistic relationship between *sdn-1* and *glp-1*, we measured *glp-1* mRNA and protein in *sdn-1(zh20)* mutant animals. First, we measured *glp-1* mRNA from extruded germlines of wild-type and *sdn-1(zh20)* animals using quantitative real-time PCR (qPCR). We found that *glp-1* mRNA levels are reduced by ~50% in *sdn-1(zh20)* mutant animals (Fig. 2b). The germline preparations used for RNA extraction contained oocytes and some embryos, both of which accumulate maternal *glp-1* RNA[27]. Therefore, to examine *glp-1* transcripts specifically in the germline, we performed single-molecule

fluorescence in situ hybridization (smFISH)[28]. Previous reports showed that *glp-1* mRNA has a graded expression in the germline where it is weakly expressed at the distal end and then gradually increases proximally[19]. We confirmed this observation using smFISH (Fig. S4). However, it was impossible to quantify the expression levels at the distal end reliably due to the weak expression and background. Therefore, we measured the smFISH signal in the pachytene region. We found that *glp-1* transcript levels in the germline are reduced by ~20% in *sdn-1(zh20)* animals (Fig. S4). In contrast to *glp-1* mRNA, GLP-1 protein is reliably detected in the distal end of the germline, where it functions to control germ cell number in the PZ, and its translation is inhibited in the proximal region[29,30]. To quantify endogenous GLP-1 protein, we utilized a CRISPR-Cas9 generated strain in which GLP-1 is endogenously tagged with the viral V5 epitope (GLP-1::V5) (Fig. 2c)[29]. The V5 tag does not detrimentally affect germline function, with GLP-1::V5 hermaphrodites having a similar number of PZ germ cells to wild-type

(Fig. S5A)[29]. Immunofluorescence detection of GLP-1::V5, using an anti-V5 antibody, showed prominent expression at the distal end of the germline and in embryos, as shown previously with anti-GLP-1 immunofluorescence (Figs. 2c and S5B)[29,31]. We quantified GLP-1::V5 expression in the distal end of the germline and found that loss of *sdn-1* reduced GLP-1::V5 protein expression (Fig. 2c, d). Together, these data suggest that SDN-1 regulates PZ germ cell number by signaling from the somatic sheath to positively regulate *glp-1* expression in the germline.

Next, we investigated how SDN-1-mediated signaling and GLP-1 expression are linked. Our previous studies using vertebrate and invertebrate models showed that the canonical role of syndecans in controlling cell behavior is to repress transient receptor potential canonical (TRPC) channel function and thereby limit cellular $Ca^{2+}$ influx[3]. We, therefore, investigated whether SDN-1 uses this mechanism to control *glp-1* expression. Three TRPC-like channels are encoded by *C. elegans*, of which only TRP-2 is expressed robustly in germline-associated tissue outside the gametes[18]. Remarkably, the introduction of the *trp-2(gk298)* mutation to *sdn-1(zh20)* animals restored wild-type GLP-1::V5 expression (Fig. 2c, d). In addition, the reduced PZ germ cell number phenotype of *sdn-1(zh20)* animals is reversed in *sdn-1(zh20); trp-2(gk298)* double-mutant animals (Fig. 2e). We have shown that SDN-1 acts in the gonadal sheath to control germ cell number in the PZ (Fig. 1), however, TRP-2 expression has only previously been observed in the nervous system[32]. We used confocal microscopy to examine the expression of the *trp-2::mCherry* transgene (gift from Kyuhyung Kim) and detected expression in the gonadal sheath (Fig. 2f). Our previous study showed that syndecans do not affect the expression of TRP channel genes[3]. In agreement with this, we found that *trp-2::mCherry* expression in the gonadal sheath is not affected by the loss of SDN-1 (Fig. 2f). We next performed rescue experiments to confirm that TRP-2 controls GLP-1::V5 expression and PZ germ cell number (Figs. 2g, h and S6). The *sdn-1(zh20); trp-2(gk298)* double mutant exhibits wild-type GLP-1::V5 expression and PZ germ cell number (Fig. 2d, e). Therefore, we restored *trp-2* expression to *sdn-1(zh20); trp-2(gk298)* animals and examined GLP-1::V5 expression and PZ germ cell number (Figs. 2g, h and S6). We found that expression of either a fosmid containing the *trp-2* genomic region or *trp-2* cDNA driven by the gonadal sheath *lim-7* promoter restored these *sdn-1* mutant phenotypes to *sdn-1(zh20); trp-2(gk298)* animals (Figs. 2g, h and S6). Together, these data reveal that the SDN-1/TRP-2 axis regulates *glp-1* expression to influence PZ germ cell proliferation from the somatic sheath.

**Control of *glp-1* transcription by a conserved promoter motif**. How does SDN-1 signaling control *glp-1* expression? We reasoned that SDN-1 regulates *glp-1* expression at the level of transcription because the reduction of *glp-1* mRNA mirrors the reduction of GLP-1 protein in *sdn-1(zh20)* mutant animals (Fig. 2). However, transcriptional mechanisms controlling the Notch receptor in *C. elegans* are unknown. To elucidate how *glp-1* is transcriptionally regulated, we surveyed *glp-1* promoters from four *Caenorhabditis* species for conserved DNA sequences and identified a highly conserved 9 base pair (bp) sequence (Fig. 3a). In the *C. elegans* promoter, the conserved sequence (TGCCACCCG) is located at the transcriptional start site, 212 bp upstream of the ATG start codon, suggesting that this motif is a binding site for transcriptional regulators[33]. Bioinformatic analysis revealed that this conserved *glp-1* promoter motif contains a consensus sequence for the transcription factor AP-2 (TFAP2) family. To interrogate the importance of this conserved motif, we introduced a 5 bp deletion within the conserved sequence (*glp-*

*1promΔ*) and examined the GLP-1::V5 expression. Using anti-V5 immunofluorescence, we found that deleting the putative TFAP2 motif reduced GLP-1::V5 levels in the germline PZ (Fig. 3b, c).

Next, we investigated the functional importance of the putative TFAP2 motif by quantifying PZ germ cell number in wild-type *glp-1::v5* and mutant *glp-1promΔ* animals. Deletion of the putative TFAP2 motif reduced the number of PZ germ cells by ~30% (Fig. 3d). Our data suggest that SDN-1 signaling controls *glp-1* transcription. If SDN-1 controls *glp-1* transcription through the putative TFAP2 motif, we would not expect to observe an additive effect when the *glp-1promΔ* mutant was combined with *sdn-1* knockdown. In congruence, *glp-1promΔ* and *glp-1promΔ; sdn-1* RNAi animals exhibit an equivalent reduction in GLP-1::V5 expression and PZ cell number (Fig. 3e, f). In contrast, *sdn-1* RNAi reduces GLP-1::V5 expression and PZ cell number in animals that harbor a wild-type *glp-1* promoter (Fig. 3e, f). Together, these data identify a highly conserved putative TFAP2 motif within the *glp-1* promoter that is important for the regulation of *glp-1* expression and germ cell fate.

We have shown that deletion of putative TFAP2 motif reduces GLP-1 expression and PZ germ cell number (Fig. 3). Next, we used CRISPR-Cas9 to mutate two bases in the putative TFAP2 binding motif of the *glp-1::v5* promoter (*glp-1prom(aa)*) that are predicted to disrupt TFAP2 binding (Fig. 4a)[34]. Animals harboring these two base-pair mutations in the *glp-1* promoter exhibited reduced GLP-1::V5 expression and PZ germ cell number to a similar extent as the motif deletion (Fig. S7A, B), supporting a potential role for TFAP2-mediated regulation of *glp-1* expression.

**APTF-2 directly regulates *glp-1* expression**. *C. elegans* encodes four TFAP2 family members (APTF-1-4), of which APTF-2 and APTF-3 have been detected in the germline[18,19]. To determine the importance of APTF-1-4 in regulating *glp-1* expression, we used RNAi to individually knock down their expression in wild-type animals for 16 h from the L4 larval stage and measured GLP-1::V5 levels in the adult germline (Fig. 4b, c). We found that *aptf-2* RNAi reduced GLP-1::V5 protein detected in the distal end of the germline (Fig. 4b, c). In addition, *aptf-2* RNAi lowered PZ germ cell number, as did an *aptf-2(qm27)* mutant that harbors a point mutation in the APTF-2 DNA binding domain (Figs. 4d and S8)[35]. In contrast, RNAi knockdown of *aptf-1* and *aptf-3/4* caused nonsignificant changes to GLP-1::V5 expression and PZ germ cell number (Fig. 4b–d).

We found that *aptf-2* RNAi knockdown did not further reduce GLP-1::V5 expression or PZ germ cell number in the germline of animals in which the TFAP2 binding site in the *glp-1* promoter is deleted (Fig. 4e, f). This suggests that APTF-2 controls *glp-1* expression and function by directly interacting with the conserved TFAP2 binding site. We tested this assertion using electrophoretic mobility shift assays (EMSAs). First, we expressed and purified APTF-2::V5-containing nuclear fractions from mammalian HEK293T cells. Next, we asked whether APTF-2::V5 could cause a shift in biotin-labeled *glp-1* promoter migration (Fig. 4g). We found that APTF-2::V5 indeed binds to the functionally-important TFAP2 motif within the *glp-1* promoter and that this interaction competes away with an unlabeled *glp-1* promoter (Figs. 4g and S9). Further, deletion of the TFAP2 motif abrogates APTF-2 binding to the *glp-1* promoter, confirming its importance in regulating *glp-1* expression (Fig. 4g). These data show that APTF-2 directly binds to the *glp-1* promoter to drive its expression and function in the *C. elegans* germline.

**APTF-2 regulation of GLP-1 requires SDN-1**. Our data show that APTF-2 positively regulates *glp-1* expression by directly binding to the *glp-1* promoter. Previous in vitro studies in mammalian

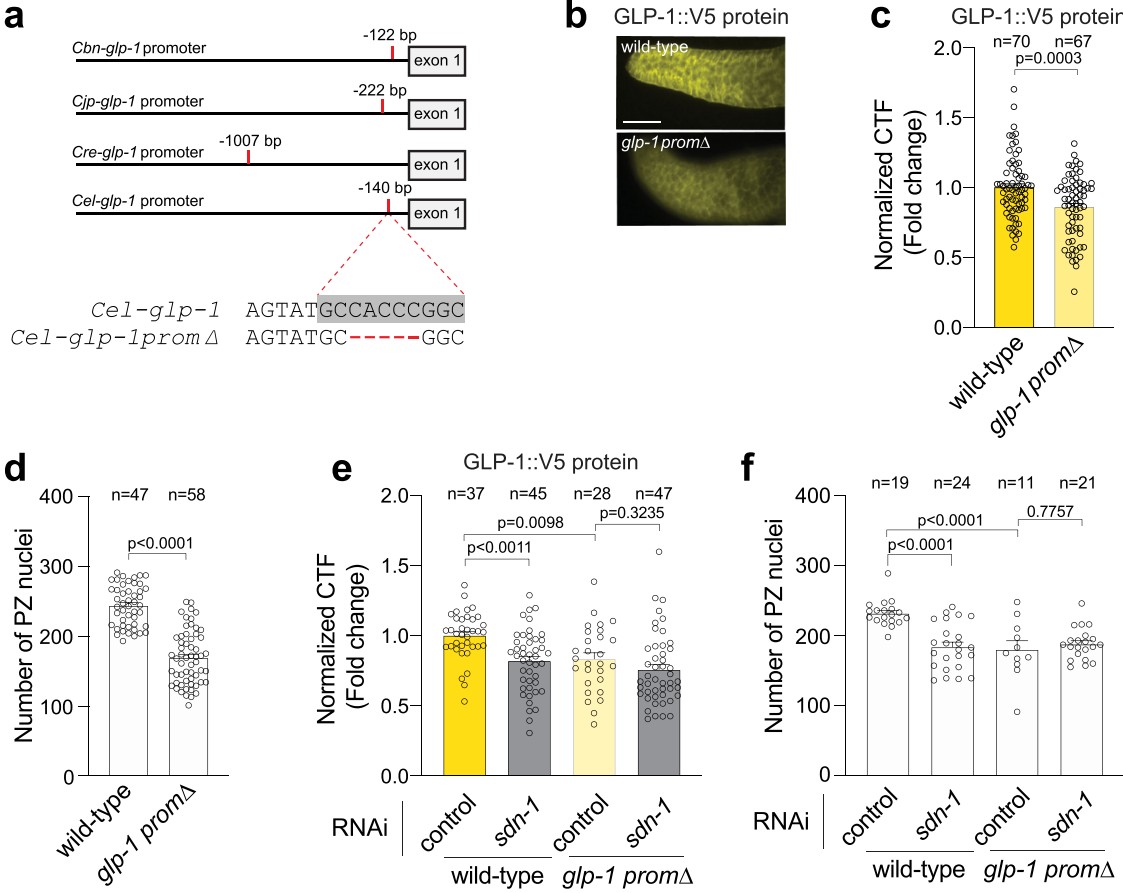

**Fig. 3 A conserved promoter motif controls GLP-1 expression and germ cell number. a** Location of a conserved motif in the *glp-1* promoter of four *Caenorhabditis* species. *Cbn C. brenneri, Cjp C. japonica, Cre C. remanei, Cel C. elegans*. The conserved motif in *C. elegans glp-1* promoter is highlighted in gray. Deleted bases in CRISPR-engineered *glp-1* promoter mutant in red. **b, c** Immunofluorescence images (**b**) and quantification (**c**) of GLP-1::V5 protein in the distal germline of wild-type and *glp-1promΔ* (5 bp deletion in conserved promoter motif) adult hermaphrodites. Data were expressed as mean ± s.e.m. and statistical significance was assessed by unpaired *t*-test. Scale bar = 10 μm. **d** Quantification of PZ nuclei of wild-type and *glp-1promΔ* adult hermaphrodites. Data were expressed as mean ± s.e.m. and statistical significance was assessed by a two-tailed unpaired *t*-test. **e** Quantification of GLP-1::V5 protein in the distal germline of adult hermaphrodites (± *sdn-1* RNAi) expressing wild-type *glp-1* or *glp-1promΔ* adult hermaphrodites. Data were expressed as mean ± s.e.m. and statistical significance was assessed by ordinary one-way ANOVA. **f** Quantification of PZ nuclei in the distal germline of adult hermaphrodites (± *sdn-1* RNAi) expressing wild-type *glp-1* or *glp-1promΔ*. Data were expressed as mean ± s.e.m. and statistical significance was assessed by ordinary one-way ANOVA. For all panels, *n* refers to the number of animals analyzed. *P* values as indicated. Source data are provided as a Source Data file.

cells showed that APTF transcriptional activity can be modulated by calcium[36]. We, therefore, performed a genetic analysis to determine whether regulation of GLP-1::V5 expression and PZ germ cell number by APTF-2 is dependent on SDN-1/TRP-2.

If SDN-1 controls PZ germ cell number through the APTF-2 regulation of GLP-1, we would expect that SDN-1 regulation of *glp-1* expression is APTF-2-dependent. To examine this, we measured GLP-1::V5 levels in wild-type and *sdn-1(zh20)* animals after RNAi knockdown of *aptf-2*. We found that GLP-1::V5 levels are reduced in *sdn-1(zh20)* animals, however, this was not further reduced by *aptf-2* RNAi knockdown (Fig. 5a). To examine the biological relevance of this finding, we performed *aptf-2* RNAi knockdown in *sdn-1(zh20)* mutant animals and analyzed PZ germ cell number. We found that unlike in wild-type animals, no reduction in PZ germ cell number was observed in *sdn-1(zh20)* mutant animals following RNAi knockdown of *aptf-2* (Fig. 5b). These data suggest that *sdn-1* acts in the same genetic pathway as *aptf-2* and *glp-1* to control PZ germ cell number.

We have shown that the reduced PZ germ cell number in *sdn-1(zh20)* animals is suppressed by removal of the TRP-2 channel (Fig. 2e). If SDN-1 controls PZ germ cell number genetically upstream of APTF-2, deleting the APTF-2 binding site in the *glp-1* promoter would abrogate the suppressive effect of TRP-2 loss on the *sdn-1(zh20)* mutant phenotype. Indeed, when the APTF-2 binding site in the *glp-1* promoter is deleted, removal of TRP-2 does not suppress the *sdn-1(zh20)* mutant reduction of PZ germ cell number (Fig. 5c). Together, these data posit that the SDN-1/TRP-2 axis controls APTF-2-dependent regulation of PZ germ cell number by GLP-1, likely through modulation of $Ca^{2+}$ levels.

**Calcium regulates APTF-2 binding to the *glp-1* promoter.** Mechanistically, changes in the level of cellular $Ca^{2+}$ have been shown to affect the subcellular localization of transcription factors as well as their affinity for gene regulatory regions[37–39]. We, therefore, examined these scenarios in the context of APTF-2 using *C. elegans* in vivo and mammalian cell-based methods.

To investigate whether SDN-1 regulates the expression of APTF-2, we used CRISPR-Cas9 to knock in the *gfp* coding sequence to the 3' end of the *aptf-2* gene (Fig. S10). APTF-2::GFP reporter expression was confirmed by Western blot and fluorescence microscopy, revealing nuclear expression in the germline, oocytes and embryos (Fig. 5d and S10). We found however that the localization of endogenous APTF-2::GFP

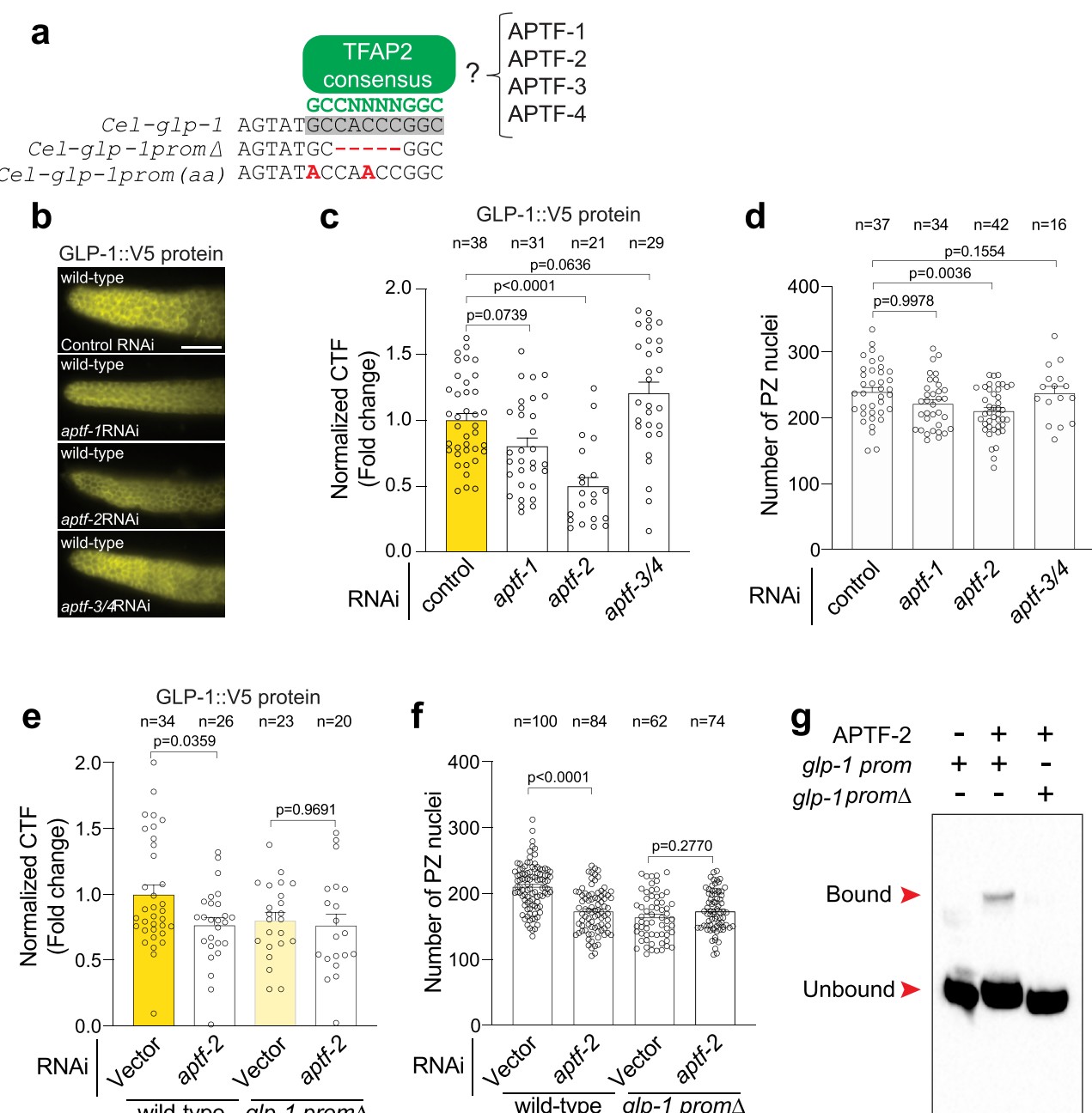

**Fig. 4 glp-1 transcription is directly regulated by APTF-2. a** A consensus binding site (GCCNNNNGGC) for AP-2 transcription factors is located within the conserved *glp-1* promoter motif. TFAP2 consensus motif (green), TFAP2 motif in the *glp-1* promoter (gray box), *glp-1* promoter deletion/mutations (red). **b**, **c** Immunofluorescence images (**b**) and quantification (**c**) of GLP-1::V5 protein in the distal germline of wild-type adult hermaphrodites following RNAi knockdown of *aptf-1*, *aptf-2*, and *aptf-3/4*. Data were expressed as mean ± s.e.m. and statistical significance was assessed by ordinary one-way ANOVA. Scale bar = 10 μm. **d** Quantification of PZ nuclei in wild-type adult hermaphrodites following RNAi knockdown of *aptf-1*, *aptf-2*, and *aptf-3/4*. Data were expressed as mean ± s.e.m. and statistical significance was assessed by ordinary one-way ANOVA. **e** Quantification of GLP-1::V5 protein in the distal germline of adult hermaphrodites (± *aptf-2* RNAi) expressing wild-type *glp-1* or *glp-1promΔ*. Data were expressed as mean ± s.e.m. and statistical significance was assessed by ordinary one-way ANOVA. **f** Quantification of PZ nuclei in the distal germline of adult hermaphrodites (± *aptf-2* RNAi) expressing wild-type *glp-1* or *glp-1promΔ*. Data were expressed as mean ± s.e.m. and statistical significance was assessed by ordinary one-way ANOVA. Scale bar = 10 μm. **g** EMSA of APTF-2::V5 protein produced in HEK293T cells and the TFAP2 consensus motif within the *glp-1* promoter. DNA bound and unbound with APTF-2::V5 is marked. Performed in triplicate. *glp-1 prom* = wild-type and *glp-1 promΔ* = TFAP2 motif deletion. For all panels, *n* refers to the number of animals analyzed. *P* values as indicated. Source data are provided as a Source Data file.

expression is unchanged in *sdn-1(zh20)* animals (Fig. 5d). Next, using mouse embryonic fibroblasts that lack syndecan-4, and thus have constitutively high levels of intracellular $Ca^{2+}$ due to dysregulation of TRPC7 channels[3], we found that APTF-2::V5 nuclear localization is not overtly affected (Fig. 5e). These data

show that high intracellular $Ca^{2+}$ does not alter *C. elegans* APTF-2 subcellular localization. Next, we used EMSA to investigate whether $Ca^{2+}$ controls the affinity of APTF-2::V5 to the *glp-1* promoter. We found that binding of APTF-2::V5 to the *glp-1* promoter, but not a control protein-DNA interaction, was

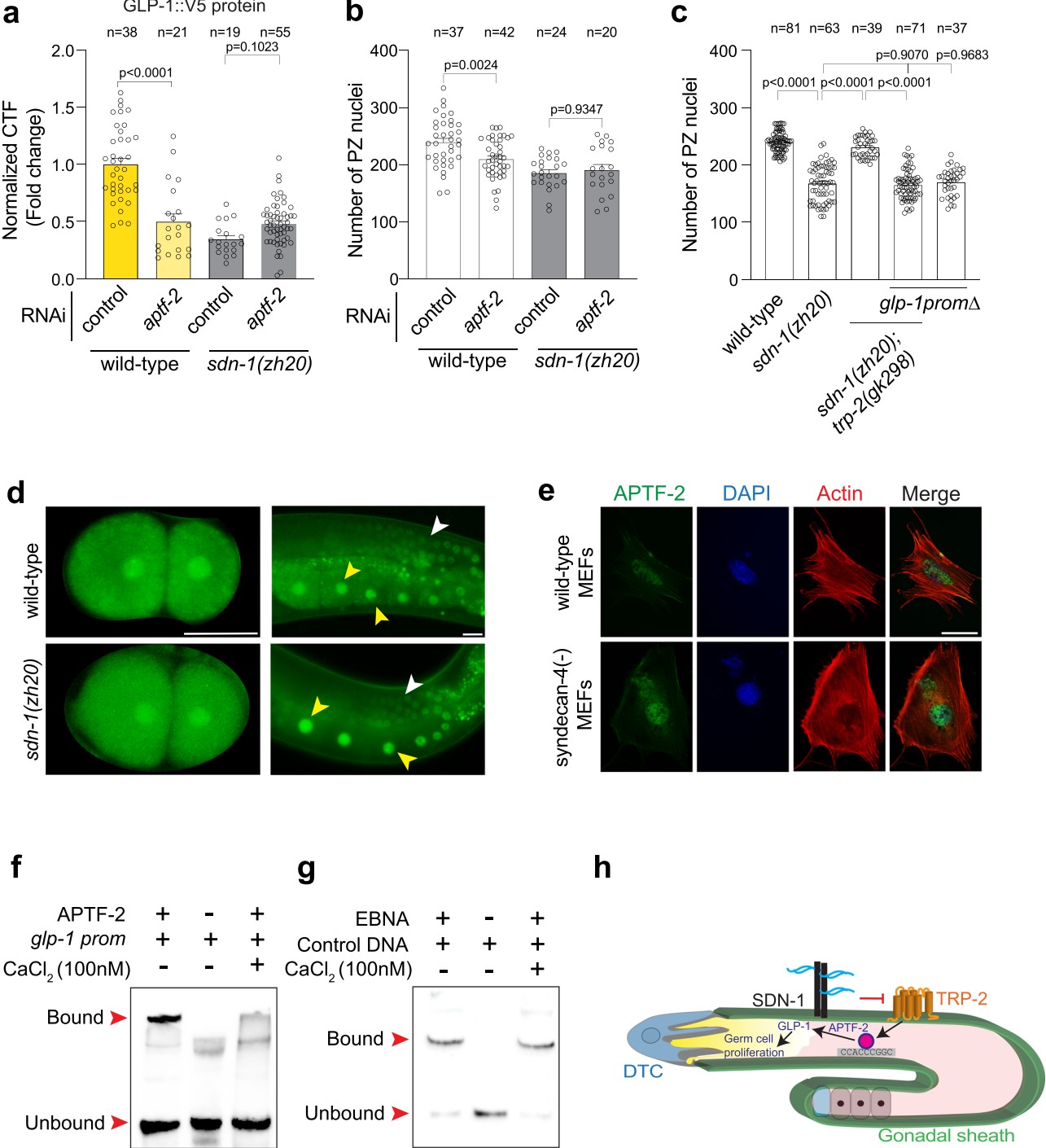

inhibited by increased Ca$^{2+}$ levels (Fig. 5f, g). Whether Ca$^{2+}$ levels directly affect the ability of APTF-2 to interact with the *glp-1* promoter or modulates an intermediate regulator of APTF-2 is yet to be discovered.

Our collective data reveal a mechanism where control of a TRP-2 channel by the transmembrane proteoglycan SDN-1 regulates GLP-1 expression and germ cell fate (Fig. 5h). Within the germline, and genetically downstream of SDN-1/TRP-2, the APTF-2 transcription factor directly regulates *glp-1* expression through a conserved TFAP2 binding site. APTF-2 binding to the *glp-1* promoter is sensitive to Ca$^{2+}$, providing a mechanistic link between the SDN-1/TRP-2 function in the gonadal sheath and *glp-1* germline transcription.

## Discussion

Here we have identified a mechanism by which the transmembrane proteoglycan SDN-1 controls germ cell fate by regulating expression of the GLP-1 Notch receptor. SDN-1 performs this function non-cell autonomously from somatic gonadal sheath cells that surround the germline. We found that the established role of SDN-1 in controlling transient receptor potential calcium channels is required for correct GLP-1 expression and germ cell fate. Further, SDN-1 requires the attachment of extracellular sugar chains to control germ cell fate, suggesting that extracellular ligands, environmental cues, or matrix proteins control this SDN-1/TRP-2 axis. In the germline, GLP-1 expression is directly regulated by the APTF-2 transcription factor through a conserved

**Fig. 5 Calcium controls APTF-2 binding to the *glp-1* promoter. a** Quantification of GLP-1::V5 protein in the distal germline of wild-type and *sdn-1(zh20)* adult hermaphrodites (± *aptf-2* RNAi). Data were expressed as mean ± s.e.m. and statistical significance was assessed by ordinary one-way ANOVA. **b** Quantification of PZ nuclei in wild-type and *sdn-1(zh20)* adult hermaphrodites (± *aptf-2* RNAi). Data were expressed as mean ± s.e.m. and statistical significance was assessed by ordinary one-way ANOVA. **c** Quantification of PZ nuclei in wild-type, *sdn-1(zh20)*, and *sdn-1(zh20); trp-2(gk298)* adult hermaphrodites expressing wild-type *glp-1* or *glp-1promΔ*. **d** Expression pattern of ATPF-2::GFP in 2-cell embryos and the adult germline of wild-type and *sdn-1(zh20)* hermaphrodites. APTF-2::GFP in the germline (white arrowheads) and oocytes (yellow arrowheads). Scale bar = 25 µm. **e** Immunofluorescence micrographs of wild-type and syndecan-4 knockout mouse embryonic fibroblasts (MEFs) expressing APTF-2::V5, co-stained with DAPI (DNA) and phalloidin (actin). APTF-2::V5 is nuclear-localized in both wild-type and syndecan-4(-) cells, which show a characteristically disorganized cytoskeleton. Performed in triplicate. >25 cells were examined. Scale bar = 50 µm. **f** EMSA of APTF-2::V5 protein produced in HEK293T cells and the TFAP2 consensus motif within the *glp-1* promoter (±100 nM CaCl₂). DNA bound and unbound with APTF-2::V5 is marked (red arrowheads). Performed in triplicate. *glp-1* prom = wild-type. **g** EMSA of Epstein-Barr Nuclear Antigen Extract (EBNA) protein extract and the control DNA duplex (±100 nM CaCl₂). DNA bound and unbound with EBNA is marked. Performed in triplicate. **h** Proposed mechanism for SDN-1 regulation of germ cell proliferation. SDN-1 inhibits the TRP-2 calcium channel in the gonadal sheath, likely controlling the influx of calcium into the germline through gap junctions. Within the germline, the calcium-dependent APTF-2 transcription factor controls *glp-1* expression by interacting with a conserved DNA sequence at the *glp-1* transcriptional start site, thereby regulating germ cell proliferation. For all panels, *n* refers to the number of animals analyzed. *P* values as indicated. Source data are provided as a Source Data file.

DNA binding motif within the *glp-1* promoter, an interaction that is dependent on the level of cellular Ca²⁺. Therefore, our cumulative data reveal a mechanism of soma-germline signaling to promote mitotic germ cell fate by regulating GLP-1 expression.

Notch signaling is an ancient and highly conserved mechanism used in metazoa to control cell-cell communication. Notch receptor mutants were first discovered in *Drosophila* and named based on the dominant notched wing phenotype observed[40]. The advent of molecular genetics enabled fine dissection of the Notch signaling pathway and its importance in development and disease[41–46]. For example, seminal discoveries revealing the function and processing of the GLP-1 and LIN-12 Notch receptors in *C. elegans* have contributed considerably to our understanding of this important signaling pathway[8,27,46–50]. In the context of the *C. elegans* germline, it has been known for decades that GLP-1 signaling intensity is important for controlling mitotic cell fate[8,14]. Posttranscriptional regulation of *glp-1* maternal mRNA is also well documented. The STAR/Quaking translational repressor GLD-1 directly binds to the *glp-1* 3′ untranslated region to limit *glp-1* mRNA levels in oocytes and at the distal end of the germline[27,30]. However, elucidation of mechanisms controlling *glp-1* transcription has been lacking. Our data show that *glp-1* is regulated through a conserved sequence located at the transcriptional start site in the *glp-1* promoter. This likely reveals an additional layer of transcriptional control of *glp-1* expression.

Previous reports have implicated proteoglycans and GAG chain modifications in the regulation and differentiation of embryonic and adult stem cells in mammals[51–54]. During embryonic stem cell (ESC) differentiation, distinct sulfation patterns in surface GAGs are observed—from low sulfation in pluripotent cells to high sulfation in differentiated cells[52,55]. Further, GAG chain-deficient ESCs are unable to differentiate, revealing a function for GAG dynamics in controlling stem cell differentiation[52]. In adult mammalian muscle, it was also shown that Syndecan-3 is required for satellite cell proliferation[53]. Intriguingly, defective cellular behavior of Syndecan-3-null satellite cells is rescued by the expression of a constitutively active Notch intracellular domain[53]. However, in this context, Syndecan-3 directly interacts with the Notch receptor and is required for Notch processing[53]. GAGs are also important for stem cell behavior in invertebrates. In *Drosophila melanogaster*, reducing 3-O sulfation causes Notch-associated neurogenic phenotypes and lower levels of Notch protein[56]. However, the mechanistic function of GAG chain sulfation and Notch signaling was not characterized. The fly model also showed that the glypican proteoglycans dally and dally-like are required for germline stem cell niche maintenance in females and males, respectively[57].

These studies in combination with our work suggest that distinct embryonic and adult stem cell decisions are controlled by proteoglycan regulation of Notch expression and processing. However, the specific regulatory modality used to control Notch may depend on evolutionary origins and tissue context.

The importance of Ca²⁺ in controlling *C. elegans* germline development has previously been shown by studies of store-operated Ca²⁺ entry (SOCE) regulators - STIM-1 (stromal interaction molecule 1) and ORAI-1 (Ca²⁺ release-activated Ca²⁺ channel protein 1). Somatic-specific RNAi knockdown of either *stim-1* or *orai-1* from early larval development causes sterility in adults, suggesting a non-cell-autonomous effect on germline development[58,59]. Our data also show that reduced germ cell proliferation caused by loss of SDN-1 is suppressed by removal of the TRP-2 Ca²⁺ channel, supporting a role for Ca²⁺ regulation of germline function. The important somatic sheath function of innexin gap junction proteins INX-8 and INX-9 in fertility suggests a possible route for communication of Ca²⁺ signals to the germline[60]. In support of this, a recent study revealed that the INX-8/9-expressing sheath cell pair 1 (Sh1) is closely associated with the majority of distal germ cells and promotes their proliferation[61]. Unfortunately, we were unable to express a calcium reporter at reliably detectable levels in either the gonadal sheath or the germline. It will be interesting to investigate in subsequent studies precisely how Ca²⁺ levels are communicated from the soma to germline and the potential function of gap junctions in this regard.

It has become clear that AP-2 transcription factors play highly conserved functions in regulating germ cell biology. Mammalian AP-2γ, encoded by the Tfap2C gene, functions with additional transcription factors (BLIMP1, PRDM14, and SOX17) to promote germ cell and pluripotency genes and to inhibit somatic fates[62–64]. Further, a recent study in the cnidarian *Hydractinia symbiolongicarpus* showed that *Tfap2*, an AP-2γ homolog, is essential for germ cell and gonad development[65]. Our data show that APTF-2, a *C. elegans* homolog of AP-2γ, promotes mitotic germ cell fate through direct, and Ca²⁺-dependent, regulation of the GLP-1 Notch receptor. ChIP-sequencing analysis in mammalian cells suggests that a direct regulatory relationship between AP-2 transcription factors and Notch is highly conserved. AP-2 transcription factors interact with the promoter of both Notch 1 and 3 in P19 embryonal carcinoma cells and with the Notch 2 promoter in MCF-7 breast cancer cells[34,66]. Therefore, further study is warranted to investigate the importance of AP-2 transcription factor regulation of Notch expression in disease processes, and how proteoglycans potentially modulate Notch transcription. To conclude, our study reveals a mechanism where

SDN-1 acts to convey somatic-germline communication to optimize germ cell behavior and progeny generation. Future work should focus on understanding how syndecans utilize their highly complex sugar chain antennae to interpret extracellular signals and optimize germ cell fates.

## Methods

**Contact for reagent and resource sharing**. Strains used in this study will be deposited at the *Caenorhabditis* Genetics Center (CGC) and will be available upon request. Further information and requests for resources and reagents should be directed to and will be fulfilled by the lead contacts Roger Pocock (roger.pocock@monash.edu) and Sandeep Gopal (sandeep.gopal@monash.edu).

### Experimental models and subject details

Caenorhabditis elegans. *C. elegans* strains used in this study were larval stage 4 or young adult hermaphrodites (as indicated in the main text or in the specific method section). *C. elegans* were cultured on Nematode Growth Medium (NGM) plates and fed with OP50 *Escherichia coli* bacteria at 20 °C unless otherwise stated. All strains used in this study are listed in Table S1. Experiments were performed in triplicates and the number of animals analyzed is annotated in each Figure legend. Ethical approval is not required for *C. elegans* studies.

*Mammalian cells.* For Electrophoretic Mobility Shift Assays, HEK293T cells were cultured in minimal essential media with 10% FCS and 2.5 mM glutamine.

### Generation of transgenic strains

sdn-1 *single copy strain.* Transgenic lines were generated by injecting DNA constructs into young adult hermaphrodites as complex arrays. *sdn-1* rescue lines were generated using miniMos-mediated genomic insertion[20]. *sdn-1(zh20)* animals were injected with a plasmid mix containing *rab-3p::mCherry::unc-54UTR*, *myo-2p::mCherry::unc-54UTR*, *myo-3p::mCherry::unc-54UTR*, *eft-3p::mos1-transposase::tbb-2UTR*, *hsp-16.41p::peel-1::tbb-2UTR* and *sdn-1::gfp::tbb-2UTR* driven by specific promoters (kind gifts of Hannes Bulow). The promoters used to drive *sdn-1* expression were *lag-2p* (DTC), *mex-5p* (germline), and *lim-7p* (gonadal sheath). Injected worms were screened for expression of fluorescent co-injection markers and positive worms were treated with 500 μl G418 antibiotic (25 μg/μl). Fourteen days after antibiotic treatment, worms were heat-shocked at 34 °C for 3 h. Surviving worms were picked individually to new plates to establish independent lines.

sdn-1p::gfp *reporter strain.* *sdn-1* endogenous promoter region (5061 bp before the start codon) was amplified from *C. elegans* genomic DNA by PCR using primers: GAAATGAAATAAGCTTTCACCTGATAGCGCGTAACG and CCAATCCCGGGGATCCATTGTCTTGTTGGTGTAATCACC. Using the In-Fusion® HD Cloning Kit (Takara Bio), the promoter was inserted into the pPD95.75 vector that had been digested with HindIII and BamHI. The sequence-confirmed plasmids were injected to wild-type *C. elegans* at 30 ng/μl concentration along with *myo-2p::mCherry* as a co-injection marker. Animals were maintained by picking animals positive for the co-injection marker.

lim-7p::gfp *reporter strain.* *lim-7* promoter was amplified by PCR from the *lim-7p::sdn-1::gfp::tbb-2UTR* (see above) plasmid using primers CGACTCTAGAGGATCAGTACTTGTGCCTTGATTCTCCTG and TCATTTTTTCTACCGCCAATCGACAGCCGGCAC. Using the In-Fusion® HD Cloning Kit (Takara Bio), the purified promoter was inserted into pPD95.75 vector that had been digested with XbaI and BamHI. The plasmid was injected into wild-type *C. elegans* at 20 ng/μl with *myo-2p::mCherry* as a co-injection marker. Animals were maintained by picking animals positive for the co-injection marker. *lim-7p::gfp; sdn-1(zh20)* worms were created by crossing *sdn-1(zh20)* to wild-type worms expressing *lim-7p::gfp*.

lim-7p::trp-2 *rescue strain.* *lim-7p::trp-2* was inserted into pPD49.26 plasmids in a two-step process. First, the pPD49.26 vector was linearized using NheI and NcoI. The *lim-7* promoter was amplified by PCR from the *lim-7p::sdn-1::gfp::tbb-2UTR* (see above) plasmid using primers AGGACCCTTGGCTAGCATGACATCATCAGAAGCATCCGAG and GATATCAATACCATGGTCATAAAAGTTCATCAGTGTCGTC. To create pPD49.26-*lim7p*, the promoter was inserted into the linearized pPD49.26 vector using the In-Fusion® HD Cloning Kit (Takara Bio). In the second step, *trp-2* cDNA was commercially synthesized (Integrated DNA technologies) with EcoRV restriction sites at both ends. The commercial plasmid containing *trp-2* cDNA was digested with EcoRV and *trp-2* cDNA was purified. The pPD49.26-*lim7p* plasmid was linearized by EcoRV and ligated to the *trp-2* cDNA. The plasmid was injected into *sdn-1(zh20); trp-2(gk298); glp-1::V5* worms at 10 ng/μl with *myo-2p::mCherry* as a co-injection marker. Animals were maintained by picking animals positive for the co-injection marker.

### Generation of strains using CRISPR-Cas9

*Endogenous tagging of APTF-2 with GFP using CRISPR-Cas9.* A C-terminal GFP knock-in strain for endogenous *aptf-2* was generated using CRISPR/Cas9-triggered homologous recombination[67]. The Cas9-sgRNA construct of *aptf-2* was obtained using PCR to insert the target sequence: (GCCAACTGAATCAAAGCCAGAGG) into pDD162 (Addgene #47549). For the GFP tag knock-in construct, homology recombination templates were constructed with the following steps: (1) cloning a 2 kb genomic region of *aptf-2* centered on the knock-in site into the pPD95.75 vector, (2) insertion of DNA encoding GFP and a linker of six amino acids (AGACCCAAGCTTGGTACC) between the last amino acid codon of *aptf-2* and the stop codon, (3) the PAM site within the recombination template was mutated to prevent Cas9 cleavage. The following mix was then injected into wild-type animals: 10 ng/μl homologous knock-in repair template, 10 ng/μl Cas9-sgRNA plasmid, *myo-2::mCherry* plasmid (4 ng/μl). Individual F1 progeny of injected wild-type worms were picked to individual plates and F2 progeny screened for GFP knock-in by PCR. After confirmation of insertion by Sanger sequencing, the GFP knock-in was outcrossed three times prior to analysis. APTF-2::GFP was imaged in 1-day-old adult germlines and two-cell stage embryos using a Zeiss Axiocam 40x objective.

glp-1::V5 *promoter mutations.* Repair oligos and injections were performed essentially as described by injecting into *glp-1::V5(q1000)* animals[68]. To generate mutations at the APTF-2 target site in the *glp-1* promoter we injected the following mix: Cas9 protein (5 μg), tracrRNA (0.4 μg/μl), crRNA (aatgggcggagtatgccacc) (0.4 μg/μl), ssDNA oligo repair donor (1 μg/μl), *myo-2::mCherry* plasmid (4 ng/μl) and brought to a 20 μl volume with nuclease-free water. Mutation/deletion of the APTF-2 binding site was screened by PCR followed by restriction digest with NciI. Mutations were verified by Sanger sequencing and outcrossed three times prior to analysis.

*Germline staining.* Semi-automated germline analysis was performed[15]. L4 hermaphrodites were picked to OP50 plates and incubated for 16 h at 20 °C to reach the young adult stage. Germlines were extruded from sedated worms and fixed on poly-L-lysine coated slides using ice-cold methanol for 1 min and then in 3.7% paraformaldehyde (PFA) for 25 min. Fixed germlines were washed three times in phosphate-buffered saline (PBS, pH 7.4) and blocked using 30% normal goat serum before incubating with primary antibodies overnight at 4 °C. After incubation, germlines were washed three times with PBS containing 1% Tween-20 (PBST) and incubated with fluorophore-conjugated secondary antibodies and 4′,6-diamidino-2-phenylindole (DAPI) for 1 h at 25 °C. After staining, germlines were washed three times with PBST. Slides were mounted by applying a drop of Fluoroshield mounting media (Sigma) on the germlines followed by a coverslip.

*Germline imaging and presentation.* Stained germlines were analyzed using either the Zeiss Axiocam 40x objective (GLP-1-V5 staining and *glp-1* smFISH) using Zen software (version 2.0.0.0 Blue) or the Leica SP5 microscope 63x objective (scoring germ cell number) with 1.7x camera zoom using Imaris 9.1.2 software. To visualize the expression of *sdn-1* in the gonadal sheath, animals expressing *sdn-1p::gfp* were imaged using the Leica SP8 microscope 63x objective. Images were converted to .tif files and cropped to show the expression. Images were converted to pseudocolors in Adobe Photoshop 2021, then cropped and organized into Figures using Adobe Illustrator 2021. When it was not possible to crop the image to a rectangle, the cropped images were placed on a rectangular black background for uniformity. A similar approach was used to image and worms expressing *lim-7p::gfp*. Worms expressing *trp-2p::mCherry* were imaged using the Zeiss 780 confocal microscope 40x objective and images were processed similarly to the *sdn-1p::gfp* images.

*Germline analysis.* GLP-1-V5 expression and *glp-1* smFISH fluorescence was quantified using FIJI according to the formula, corrected total fluorescence (CTF) = integrated density – (area of the mitotic region × mean fluorescence of background readings). The CTCF values are normalized against wild-type or controls of each experiment so that intensity changes occurred due to microscope and light source conditions between experiments can be removed.

Germ cell number and 3D modeling was performed using Imaris 9.5[15]. First, the confocal images were converted to Imaris files. The mitotic region was defined manually based on nuclei shape. The proximal end of the proliferation zone was defined as immediately prior to detection of two crescent-shaped nuclei, marking the start of the transition zone. Each Z-plane within the mitotic region was selected using marking tools within the Imaris software. Nuclear size within the manually defined mitotic region was set at 2.0 μm under the current imaging conditions. The defined nuclei size varies according to the objective, laser-quality, magnification, and camera parameters. Based on the defined size, the nuclei were resolved into 3D models. The method was validated by visually identifying the nuclei and scoring the cell number using wild-type germlines. To quantify the number of cells in active mitosis, extruded germlines were stained with an anti-pH3 antibody (Rabbit-monoclonal anti-pH3 Sigma Aldrich 06570 (clone H3S10P) - 1:1000) and pH3-positive nuclei counted.

*Transfection of HEK293T cells.* HEK293T cells were seeded on a 3 cm dish 1 day prior to transfection. Cells were transfected at 80% confluency using Lipofectamine

2000 (Thermofisher) Briefly, 2 µg pcDNA3.2-aptf2::V5 was diluted in 200 µl Opti-MEM (Life Technologies) and 8 µl Lipofectamine was diluted in 200 µl Opti-MEM. Both solutions were incubated separately at 25 °C for 5 min before being combined for further incubation at 25 °C for 20 min. The mixture was added to the dish containing cells in 1 ml of media and incubated for 24 h at 37 °C prior to harvesting for nuclear preparation and Electrophoretic Mobility Shift Assay.

*Biotin labeling and DNA duplex formation.* 60 bp oligonucleotides encompassing part of the *glp-1* promoter (wild-type or with the TFAP2 consensus sequence deleted - AP-2 del) (Table S1) were labeled using a 3′ end DNA biotin labeling kit (Thermo Scientific). First, 1 µM oligonucleotides were incubated with 0.12 U/µl terminal deoxynucleotidyl transferase, 1x terminal deoxynucleotidyl transferase buffer (500 mM cacodylic acid, 10 mM $CoCl_2$, 1 mM DTT, pH 7.2), and 0.5 µM Biotin-11-UTP at 37 °C for 30 min. The reaction was stopped by adding 2.5 µl of 0.2 M EDTA and labeled oligonucleotides extracted from the reaction mix by adding 50 µl chloroform:isoamyl alcohol. Biotin-labeled complementary oligonucleotides were incubated together at 90 °C for 2 min. The temperature was gradually reduced to 70 °C and incubated for 30 min before further reducing the temperature to 10 °C. This resulted in the formation of DNA duplexes with biotin labels at the 3′ ends.

*Electrophoretic mobility shift assay (EMSA).* HEK293T cells (see above) were washed with PBS and nuclear fractions extracted using NE-PER Nuclear and Cytoplasmic Extraction Reagents (Thermo Scientific). Briefly, 50 µl of the packed volume of cells was lysed with 500 µl of cytoplasmic extraction reagent I and 27.5 µl of cytoplasmic extraction reagent II to remove the cytoplasmic fraction. The remaining cell pellets were treated with 500 µl of nuclear extraction reagents for 40 min and the solution was centrifuged at 20,893x*g* for 5 min. The supernatant containing the nuclear fraction was collected for EMSA. EMSA was performed according to the instructions from the LightShift® Chemiluminescent EMSA Kit (Thermo Scientific). Eight microliters of the nuclear fraction was incubated with 15 nM of biotin-labeled DNA duplex, 2.5% glycerol, 5 mM $MgCl_2$, 0.05% NP-40, and 50 ng/µl Poly(deoxyinosinic-deoxycytidylic) acid for 25 min at 25 °C. Three sets of reactions were prepared: (1) APTF-2::V5 nuclear fraction and biotin-labeled wild-type *glp-1* DNA duplex, (2) APTF-2::V5 nuclear fraction and biotin-labeled *glp-1* AP-2 del DNA duplex, and (3) biotin-labeled wild-type *glp-1* DNA duplex alone. Another set of competition reaction mixes were prepared by adding incremental amounts of unlabeled wild-type *glp-1* DNA duplex in addition to biotin-labeled wild-type *glp-1* DNA duplex. For all reactions, an equal amount of APTF-2 nuclear fraction was used. After a 30 min incubation, 5 µl of 5x loading buffer was added to the reaction mix. Samples were then run on a native polyacrylamide gel in 1x Tris-Borate-EDTA buffer. Gels were blotted onto nitrocellulose membranes. The membranes containing DNA were crosslinked using a UV transilluminator for 15 min, followed by incubation with blocking buffer provided in the EMSA kit for 15 min at 25 °C. Then, the membrane was incubated with blocking buffer containing stabilized Streptavidin-Horseradish Peroxidase Conjugate (1:300) for 15 min followed by five washes with wash buffer. The membrane was then incubated in substrate equilibration buffer for 15 min and treated with a chemiluminescent substrate for 5 min. The membrane was exposed using Biorad ChemiDoc XRS + and images were taken using a CCD camera and Image lab 6.0.1 software. To examine the effect of $Ca^{2+}$ on APTF-2 binding to the *glp-1* promoter, the EMSA was performed with the reaction mix containing 100 nM of $CaCl_2$. Controls were performed by using a reaction mix containing Epstein-Barr Nuclear Antigen Extract (1U) and 60 bp biotin end-labeled duplex (20 fmol) containing the following binding site 5′-TAGCATATGCTA-3′.

*Single-molecule fluorescence in situ hybridization (smFISH).* smFISH was performed using Stellaris RNAi FISH reagents (Biosearch Technologies) on both the germline and embryos. To perform smFISH, germlines were dissected from sedated worms and placed on slides before being fixed using 3.7% PFA in PBS for 45 min at 25 °C. Germlines were then permeabilized using 70% ethanol at 4 °C for 2 h followed by incubation with Buffer A at 25 °C for 5 min. The *glp-1* FISH probes were prepared in 100 µl of Hybridization Buffer and germlines were incubated in the probe solution in the dark at 37 °C for 4 h. A total of 500 µl Buffer A was added to the germline and incubated at 37 °C for 30 min followed by washing with Buffer A. Samples were then incubated with DAPI stain solution (Wash Buffer A consisting of 5 ng/ml DAPI). After a 30 min incubation in the dark at 37 °C, the DAPI solution was removed, and germlines were washed with Buffer B. Images were captured using the Zeiss Axiocam 40x objective. The expression of *glp-1 mRNA* gradually increases from the distal to the proximal end. The distal end is nearly devoid of *glp-1* mRNA. Therefore, the distal-most end of the germline was used as the background when quantifying *glp-1* mRNA. Three areas of the same size were selected in the region immediately after the transition zone and the CTF of these areas were calculated using the formula above (Fig. S4).

*RNA interference.* HT115(DE3) *E. coli* bacteria expressing RNAi plasmids for specific genes or empty vector (L4440) were grown in Luria Broth (LB) + Ampicillin (50 µg/ml) at 37 °C for 16 h. Saturated cultures of RNAi bacteria were plated on RNAi plates and allowed to dry for 24 h. L4 hermaphrodites were placed on the RNAi plates and incubated for 16 h at 20 °C before proceeding with germline analysis.

*Brood size analysis.* Ten L4 hermaphrodites were picked onto individual NGM plates seeded with OP50 bacteria. Worms were allowed to lay eggs for 24 h and then the mothers were individually moved to new plates. After a further 24 h embryos were analyzed for hatching. This process was repeated for 6 days. The number of larvae and embryos were counted each day and summed as the total brood size.

*Western blotting.* Transgenic worms expressing GFP-tagged APTF-2 were grown on NGM plates coated with OP50 bacteria. A packed volume of 200 µl of mixed stage worms and embryos were collected by washing with M9 buffer. Worms were washed three times with M9 buffer, pelleted by centrifugation, and 250 µl of lysis buffer (50 mM Tris pH 7.4, 150 mM NaCl, 2% Triton X-100, 0.1% SDS, 1x protease inhibitor cocktail) was added. The samples were sonicated using Bioruptor® (Diagenode) at high amplitude (4 °C; 30 s ON and 20 s OFF) for 10–15 cycles. Worms were then disrupted using a mortar and pestle for 5 min before centrifuging at 735×*g* for 5 min. The supernatant was collected, and LDS sample buffer (1x final concentration) was added before boiling at 95 °C for 10 min. Samples were cooled to room temperature and run on a 10% polyacrylamide gel. The gel was blotted to a PVDF membrane using the iBlot semi-dry blot system (Thermo Scientific). After blocking with 5% BSA in PBS for 1 h, the PVDF membranes were incubated with anti-GFP antibody (Mouse-polyclonal anti-GFP Sigma Aldrich 11814460001 - 1:1000) and α-tubulin (Mouse monoclonal anti-alpha-tubulin (clone 12G10) Developmental Studies Hybridoma Bank AB1157911 - 1:200) in 1% BSA in PBS-tween (0.1%) for 16 h at 4 °C. The membrane was then washed three times with PBS-tween and incubated with horseradish peroxidase-conjugated secondary antibodies in 1% BSA in PBS-tween for 1 h at 22 °C. The membrane was washed five times with PBS-tween and incubated with ECL reagents (Thermo Scientific) for 5 min at 22 °C. The membrane was exposed using Biorad ChemiDoc XRS + and images taken using a CCD camera and Image lab 6.0.1 software.

*cDNA preparation and quantitative PCR.* About 200 L4 stage hermaphrodites were grown on NGM plates for 16 h, and the adult germlines were extruded. Total RNA from extruded germlines were isolated using the Qiagen RNAeasy kit. First, germlines were placed in 350 µl of lysis buffer and snap-frozen in liquid nitrogen before being thawed at 37 °C. Freeze/thaw cycles were repeated seven times and an equal volume of 70% ethanol was added. The samples were transferred to RNAeasy spin columns and centrifuged at 13793 × *g* for 15 s to remove flow through. To eliminate DNA contamination, on-column DNA digestion was performed by adding DNase in RDD buffer. Samples were then washed twice with wash buffers followed by 80% ethanol before being eluted in 14 µl of RNAase-free water. cDNA was synthesized from the total RNA using the ImProm-II™ Reverse Transcription System. Briefly, 500 ng of RNA was mixed with 5 µl of oligo-dT (0.05 µg/µl) at 70 °C for 5 min and then 15 µl of master mix (1x ImProm II buffer, 0.05 mM dNTP, 20 U RNasin Ribonuclease inhibitor, 2.5 mM $MgCl_2$ and 1 U ImProm II) was added. cDNA was synthesized in a thermocycler (5 min at 25 °C, 1 h at 42 °C, 15 min at 70 °C). To perform qPCR, 1 µl of cDNA was mixed with 5 µl of SyBR master mix (Roche), 1 µl of forward primer (10 µM) and 1 µl reverse primer (10 µM) on 96-well plates. Reactions were run on a LightCycler 480 II real-time PCR machine for 35 cycles. Two housekeeping genes, *cdc-42* and *pmp-3*, were used as reference gene controls. The Ct values obtained were used to calculate the mRNA levels and values were normalized against values obtained from the reference genes. Graphs were plotted as a fold change of *glp-1* mRNA in *sdn-1* mutants with respect to the wild-type worms.

*Site-directed mutagenesis for SDN-1 construct (S-A mutations).* Three heparan sulfate chains attach to the SDN-1 core protein at [71]Ser, [86]Ser, and [214]Ser. To prevent heparan sulfate chain attachment to the core protein, these serine residues were substituted with alanine using the Q5® Site-Directed Mutagenesis Kit. [71]Ser and [86]Ser were substituted in a single reaction using the oligos oSA1F and oSA1R (Table S1). The plasmid containing *lim-7p::sdn-1::gfp::tbb-2UTR* was used as the template for the first two substitutions. The resulting plasmid after the first substitution was used for mutating [214]Ser, using the primers oSA2F and oSA2R (Table S1). For both mutations, a reaction mix containing 25 ng of template DNA, 1x Q5 Hot Start High-Fidelity 2X Master Mix, 0.5 µM forward primer and 0.5 µM reverse primer was prepared and run on a thermocycler. After initial denaturation at 98 °C for 30 s, 25 cycles of 98 °C for 10 sec, 62 °C for 20 s, and 72 °C for 5 min were run. The reaction ended with a final extension for 2 min at 72 °C. The new plasmid was then transformed, purified, and the substitutions were confirmed by Sanger sequencing.

*APTF-2-V5 expression mouse embryonic fibroblasts.* Wild-type and syndecan-4 knockout mouse embryonic fibroblasts (MEFs) were transfected with APTF-2::V5 as explained above. Cells were incubated for 48 h after transfection at 37 °C before fixing with 4% PFA in PBS. Cells were then incubated with anti-V5 antibody (Mouse monoclonal anti-V5 (clone SV5-pk1) BioRad MCA1630 - 1:1000) at 4 °C overnight before incubating with fluorophore-tagged secondary antibody, DAPI and phalloidin. Cells were imaged using the Zeiss Axiocam 40x objective.

*Quantification and statistical analysis.* All experiments were performed in three independent replicates and the experimenter was blinded to genotype. Statistical

analysis was performed in GraphPad Prism 7 using one-way analysis of variance (ANOVA) for comparison followed by Dunnett's multiple comparison test where applicable. An unpaired $t$-test was performed if the comparison was for two conditions. Values are expressed as mean ± s.e. Differences with a $p$ value $< 0.05$ were considered significant.

**Reporting Summary**. Further information on research design is available in the Nature Research Reporting Summary linked to this article.

## Data availability

All data is available in the main text or supplementary materials. In addition, Source Data are provided with this paper. There are no accession codes, unique identifiers, or weblinks in our study and no restrictions on data availability. Source data are provided with this paper.

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

## Acknowledgements

We thank Sarah Crittenden, Hannes Bulow, Alicia Melendez, and John Couchman and members of the Pocock Laboratory and for advice and comments on the manuscript. Some strains were provided by the *Caenorhabditis* Genetics Center (University of Minnesota), which is funded by the NIH Office of Research Infrastructure Programs (P40 OD010440). We extend our thanks Guangshou Ou and Zhiwen Zu for their expertise and advice in genome-editing to generate CRISPR/Cas9 tagged APTF-2::GFP. We also appreciate Judith Kimble and Sarah Crittenden for making the GLP-1::V5 CRISPR/Cas9 strain available prior to publication, for the kind gift of *sdn-1* Minimos rescue constructs by Hannes Bulow and Dayse S. da Cunha, and Monash MicroImaging for their support. We also thank Kyuhyung Kim for the *trp-2p::mCherry* transgene. This work was supported by the following grants: NHMRC (GNT1105374 and GNT1137645 to R.P.; GNT1161439 to S.G.), ARC (DP200103293 to R.P. and DE190100174 to S.G.), and veski Innovation Fellowship (VIF23 to R.P.).

## Author contributions

Conceptualization, S.G. and R.P.; Methodology, S.G. and R.P.; Investigation, S.G., A.A., A.E., L.N., and R.P. Writing—Original Draft, R.P.; Writing—Review and Editing, S.G., A.A., L.N. and R.P.; Funding acquisition, S.G. and R.P.; Resources, S.G. and R.P.; Supervision, S.G. and R.P.

## Competing interests

The authors declare no competing interests.
