## [Peer Review File · Nature Communications]

REVIEWER COMMENTS

Reviewer #1 (Remarks to the Author):

Summary:

Gopal and co-authors present convincing and very elegant evidence implicating Syndecan in the transcriptional control of Notch receptor expression to regulate germ cell number in the PZ/mitotic zone of the germline in *C.elegans*. Mutants of Syndecan (2 independent mutations), post-developmental depletion of Syndecan by RNA interference, and rescue of the germline defects with single-copy insertion of Syndecan expressed in the somatic gonad sheath cells, all conclusively establish a role for Syndecan in this context. Importantly, double loss of Syndecan and Notch receptor has similar consequences as the loss of each of the single gene losses, suggesting that Syndecan and the Notch receptor may function in the same pathway. Notch receptor expression level, both mRNA and protein, depends on functional Syndecan.

To define the mechanism regulating germ cell number, the authors successfully identify a novel Notch receptor transcriptional regulatory sequence (TFPA2 motif), and find that the germline expressed transcription factor APTF-2 is required for regulating Notch receptor expression and germ cell number in vivo (using *aptf-2* RNA interference and mutants), as well as competitively binding the TFPA2 motif of the Notch receptor locus in biochemical assays. Deletion of the TFAP2 motif does not enhance the *aptf-2* depletion defects, consistent with a direct regulation, which the authors confirm by EMSA. Further, the authors find that APTF-2 regulation of the Notch receptor requires Syndecan. Finally, the authors show that Ca²⁺ regulates APTF-2 Binding to the *glp-1* promoter in biochemical assays. Given that in other contexts Syndecan can impact cell behavior by repressing transient receptor potential canonical (TRPC) channel function, resulting in limited cellular Ca²⁺ influx, the authors explore a role for this mode of action of Syndecan (via TRPC) in the control of proliferation in the germline, by studying the sole TRP channel expressed in the gonad of *C.elegans*, TRP2. The authors find that mutation of TRP2 suppresses the reduced levels of GLP-1::V5 expression levels and PZ germ cell number observed in Syndecan mutants, implicating TRP2 in this context, which is suggestive that Syndecan effects could be through Ca²⁺ levels in this biological context as well.

The authors also offer evidence linking the regulatory role of Syndecan, potentially via the gonad-expressed Ca²⁺ channel TRP2, and propose that Syndecan's impact on TRP2 affects Ca²⁺ levels, resulting in changes on transcriptional activity of APTF-2. Since APTF-2 transcriptional activity has been found to be regulated by Ca²⁺ in other contexts, the authors test if regulation of Notch receptor expression (GLP-1::V5) and PZ germ cell number by APTF-2 depends on SDN-1/TRP2. They measure GLP-1::V5 levels in wild-type and Syndecan mutants upon *aptf-2* reduction by RNA interference, which did not further reduce Notch receptor levels compared to the low levels present in Syndecan mutants, in agreement with the possibility that Syndecan acts in the same genetic pathway as *aptf-2* and *glp-1* to control PZ germ cell number.

The authors further elegantly demonstrate that suppression of *sdn-1* germ cell number defect by loss of TRP2 is dependent on the binding of APTF-2 to *glp-1* promoter (as deletion of APTF2 binding site in the *glp-1* promoter prevents suppression of *sdn-1* by TRP2 mutation). This is suggestive that modulation by SDN-1-TRP2 may impact APTF-2 regulation of *glp-1* Notch receptor.

Evaluation:

Overall, this is an excellent and exciting piece of work. The experiments are carried out with great care, using state-of-the art approaches, powerfully combining in vivo molecular genetics with biochemical and cell biological approaches. The manuscript is well written and the figures are well crafted. The work solidly demonstrates that the proteoglycan Syndecan impacts Notch receptor expression in the germline, and that an AP-2 transcription factor (APTF-2) to the *glp-1* promoter is the mediator of *glp-1* transcriptional regulation. While functional links between Syndecan and Notch signaling have been previously reported, the definition of transcriptional control elements/ pathway for the Notch receptor, implicating the proteoglycan Syndecan, is of great importance to the field of stem cells and Notch. The link between Syndecan and transcription of *glp-1*/Notch receptor is proposed to be via Ca²⁺ levels (as APTF-2 depends on Ca²⁺ for its binding to the promoter of *glp-1*), and would be

affected by the somatic TRP calcium channel, be regulated by Syndecan. The demonstration for this part of the pathway is less strong.

Key points of the mechanism remain unaddressed, and the only alternative discussed is that TRP2 and Syndecan are functioning in the same cells, ie., the somatic gonad. The expression and localization of TRP2 is not investigated (besides citing that it is expressed in the somatic gonad), Ca²⁺ levels in the germline are not directly assayed (in Syndecan or TRP2 mutants, or double), and the localization of Syndecan is not explored (it functions from the somatic gonadal sheath cells as its expression in/from these cells suffices for function, but does Syndecan remain at the somatic cell surface, or is it shed into the germline to directly act on the germ cell nuclei?

A few experiments (especially points 1-3 below) would strengthen the conclusions and improve the manuscript; other issues could be clarified by text changes.

1) Concerning the mechanism by which Syndecan may impact Notch receptor expression, via TRP2/Ca²⁺. Syndecan function impacting germline is recapitulated by expression in gonadal sheath alone. In the Results summary paragraph, it is claimed that "control of a TRP2 channel by the transmembrane proteoglycan SDN-1 regulates GLP-1 expression", and the assumption of Syndecan acting at the gonad membrane remains (e.g., in first paragraph of Discussion and in Abstract). It may be that Syndecan indeed remains transmembrane bound at the somatic cell surface, yet there is no investigation of this. An alternative is that Syndecan may be shed from the gonad sheath cells (as it is in numerous contexts). It is conceivable that, once shed, Syndecan could directly act on germ cells or on the syncytial plasma membrane surrounding germ cells, and that the effect of TRP2 may be through regulating the shedding of Syndecan. Syndecan shedding/localization could be investigated by visualizing the N-terminus and the C-terminus of Syndecan with different fluorescent labels. Related to this, what precise somatic gonad cells express TRP2 and where is it required to function to impact APTF2-mediated Notch receptor transcriptional expression? Can TRP2 mutant defects be rescued by expression under the same driver as used to rescue Syndecan mutants (gonadal sheath = *lim-7* promoter)? ie, do Syndecan and TRP2 function in the same cells? Is TRP2 expression affected by Syndecan mutation?

A more basic control here is to ask whether gonad sheath cells integrity is altered in Syndecan mutants? Images show that gonad diameter is smaller and that PZ zone length is reduced (Fig S1). Is somatic gonadal sheath cell morphology (or their processes) altered in Syndecan mutant? Using a fluorescent driver to visualize the gonadal sheath cell in Syndecan mutants would address this important issue.

Syndecan expression in DTC does not rescue, which may be surprising as it can function non-autonomously. A comment would be helpful (this is not so important.)

2) About the mechanistic relationship between Syndecan and the Notch receptor: The authors have shown that Notch receptor levels (mRNA and protein) are affected by mutation of Syndecan, and define a transcriptional pathway. However, whether Syndecan mutation affects the activity of the Notch receptor has not been addressed. This seems particularly important since Syndecan is known to impact receptor activity in other contexts.

The authors could express *glp-1*/Notch receptor (using a germ line specific driver) to test for suppression of the germ cell defects in Syndecan mutant *sdn-1(zh20)*. Such overexpression of Notch receptor bypassing the need for Syndecan would provide evidence that the Notch receptor acts downstream of Syndecan, and that Syndecan affects the germ cells fate by modulating *glp-1* expression and not its activity.

3) That mutation of TRP2 suppresses the reduced levels of GLP-1::V5 expression levels and PZ germ cell number observed in Syndecan mutants suggests that Syndecan effects may be through Ca²⁺ levels given its role on Ca²⁺ in other contexts.

Additional independent evidence for a role for Ca²⁺ regulation in germ cell development is necessary. To assess Ca²⁺ levels directly, perhaps a similar Ca²⁺ sensors to that used by this group in *C.elegans* in Gopal et al, J. Cell Biol 2015 could strengthen the link with Ca²⁺ per se in this context? Alternatively, optogenetic manipulation of a Ca²⁺ channel may be a way to directly show an impact on Notch receptor transcription? Related to this, in Fig 5 F, EMSA assays are used to show Ca²⁺ dependency of for APTF2 binding to the glp-1 promoter. Is this Ca²⁺ effect on APTF-2 binding to the glp-1 promoter dose-dependent? Testing whether *inx-8* and *inx-9* indeed allow for germline Ca²⁺ levels regulation by gonadal sheath cells would add context to the mechanism.

4) The double loss of Syndecan and Notch receptor have similar consequences as the single gene losses, arguing that Syndecan and the Notch receptor function in the same pathway. Authors should comment on the nature of the Notch receptor allele *glp-1(e2141)* loss-of-function used here (null, partial?), as it is key for data interpretation.

5) That *aptf-2* reduction by RNA interference did not further reduce Notch receptor levels compared to the low levels present in Syndecan mutants, is consistent with Syndecan acting in the same genetic pathway as *aptf-2* and *glp-1* to control PZ germ cell number. However, testing the triple depletion in this context (*aptf-2* RNAi in *sdn-1*; *glp-1* double mutants, which the authors use earlier in the manuscript) would be important to conclude about the pathway.

6) The model figure depicts only molecules without any cellular context. Reusing the diagram of Fig-1e would help contextualize the genes and molecules implicated.

7) Fig. 1e should include a representation of the gonadal sheath processes onto the proximal part of the PZ (authors write "gonadal sheath consists of somatic cells that cover the surface of the germline meiotic region and also send processes into the proximal part of the PZ 22-24)

8) Syndecan HS chains appear necessary for Syndecan function on controlling germ cells. It would have been interesting to test if HS chemical modification are also involved in the process. Authors do mention in the Discussion that "Drosophila melanogaster, reducing 3-O sulfation causes Notch-associated neurogenic phenotypes and lower levels of Notch protein". It could have been tested using well established mutations of *C.elegans* or discuss a potential impact in Discussion.

9) Related to this, the authors write "Further, SDN-1 requires the attachment of extracellular sugar chains to control germ cell fate, suggesting that extracellular ligands, environmental cues or matrix proteins control this SDN-1/TRP-2 axis." An alternative possibility is that Syndecan devoid of HS chains may not be properly localised/shed?

10) This sentence is a bit confusing: "We found that unlike in wildtype animals, no reduction in PZ germ cell number was observed in *sdn-1(zh20)* mutant animals following (Figure 5B)." Maybe reword to something like "We found that RNAi knockdown of *aptf-2*, which in wildtype reduces PZ germ cell number, did not further reduce PZ germ cell number in *sdn-1* mutant animals"

11) Incomplete sentence at page 9 ? "Whereas, *sdn-1* RNAi reduces GLP-1::V5 expression and PZ cell number in animals that harbor a wild-type *glp-1* promoter (Figure 3E-F)."

12) The PZ cell number scale changes between graphs and figures, would be easier to compare results if kept constant across all figures (max= 400)?

Reviewer #2 (Remarks to the Author):

Review of "A Somatic Proteoglycan Controls Notch-Directed Germ Cell Fate"

Overview

Gopal and colleagues report a series of experiments to propose that *C. elegans* syndecans, SDN-1, promotes germline proliferation non-autonomously. They provide evidence that SDN-1 expressed in the somatic gonad regulates a Ca^{++} channel, and that changes in Ca^{++} influence binding of the APTF-2 transcription factor and its transcriptional activation of the *glp-1* gene, which encodes a Notch receptor responsible for maintenance of germline stem cells. The various lines of data provided in their manuscript support each of the steps of this model, and their data is strong for the most part and the manuscript is well written. Some assertions in the manuscript are overly strong and should either be softened or bolstered with additional data. Nonetheless, syndecans regulation of Notch expression is an exciting advance and will be of broad interest.

Major Comments

1. The manuscript would be greatly improved with a clear statement or model figure explaining how SDN-1 in somatic sheath cells affect GLP-1 expression in distal germ cells. The most distal sheath cell processes have marginal contact with distal-most germ cells in the progenitor zone. Where is SDN-1 affecting the calcium change and how is that calcium affecting germ cells in the progenitor zone? Is the idea that calcium diffuses through the syncytium? Also, are there any effects on the more proximal germ cells that have extensive contact with the sheath?

2. The manuscript reports SDN-1 regulation of *glp-1* expression and states that it occurs at the transcriptional level. For example, the Abstract states: "Hence, SDN-1 signaling promotes GLP-1 transcription and mitotic germ cell fate." Evidence for SDN-1 regulation of *glp-1* expression is compelling but evidence that it occurs the transcriptional level is suggestive rather than compelling. Moreover, the well-established post-transcriptional *glp-1* regulation is not mentioned but should be. Below are lines of evidence used to argue for transcriptional control, and the reasons they are not definitive.

(1) Argument: APTF-2 homologs are well established transcription factors and hence imply transcriptional regulation. Caveat: Some canonical transcription factors also bind RNA and regulate RNA activity, probably at the level of stability or translation; most others have not yet been investigated in this light but they cannot be assumed to be limited to transcriptional regulation.

(2) Argument: A *glp-1* promoter mutation that deletes an APTF-2 binding site reduces *glp-1* expression. Caveat: The location of the binding site relative to the transcription start site or the 5' end of the 5'UTR should be mentioned to bolster the argument that the binding site lies in the promoter. Is it possible that the binding site resides within the 5'UTR or in the nascent transcript that has not yet been trans-spliced? These may seem like ad hoc considerations, but given the prevalence of post-transcriptional regulation in the *C. elegans* progenitor zone, they are worth considering.

(3) Argument: APTF-2 binds the consensus site in the *glp-1* promoter in vitro. Caveat: In vitro binding of a DNA binding protein to its consensus site is not surprising, but it does not mean that APTF-2 binds that site in the *C. elegans* germline? Can the authors find evidence for in vivo binding in available Chip-Seq data? Although such data would not address APTF-2 action in the progenitor zone, it would bolster the argument.

(4) Argument: Using qPCR to test RNA extracted from *sdn-1* mutants and wildtype worms, *glp-1* mRNA levels were lower in the mutants. Caveat: The vast majority of *glp-1* mRNA is in oocytes, where it accumulates as maternal RNA for deposition in embryos; much less is present in the progenitor zone. This qPCR test is therefore unlikely to measure expression changes in the progenitor zone.

(5) Argument: Expression was tested with single molecule FISH to visualize *glp-1* RNAs and with immunostaining to visualize GLP-1 protein cells in wildtype and *glp-1* "promoter" mutant animals and embryos. *glp-1* mRNA was lower in mutant embryos but could not be tested in the progenitor zone due to background; GLP-1 protein was lower in the mutant progenitor zone. The lowered GLP-1 protein does indeed show an effect on expression but not on transcription. Caveat 1: The rigorous way to assay transcription in the progenitor zone is to visualize active transcription sites (ATS) in progenitor zone nuclei and to determine numbers of ATS and number of nascent transcripts at ATS. Effects on the abundance of cytoplasmic transcripts may reflect post-transcriptional rather than transcriptional regulation. ATS analyses in the progenitor zone are well established with smFISH and a

publicly available MATLAB code. It was not clear from the Methods section if smFISH was performed in dissected gonads or whole worms. The standard method uses dissected gonads. Caveat 2: *glp-1* mRNA in embryos is maternal and its differential expression is known to be subject to post-transcriptional regulators. Although differences were reported in the abundance of those maternal RNAs, those differences may not reflect transcriptional control.

Recommendation: The authors could either do additional experiments and measure transcription directly in the progenitor zone (ATS analyses) or soften their conclusion with respect to transcriptional control. The latter strategy would require revisions throughout the text but no new experiments. Their current data clearly show that SDN-1 and the APTF-2 binding site influence *glp-1* expression and this is very interesting. The simplest explanation is transcriptional control but it is not a slam dunk. We recommend that the smFISH experiments be repeated in dissected gonads to report cytoplasmic mRNAs in the progenitor zone without background, which would be a useful addition though not as definitive as an ATS analysis. The ATS analysis is more challenging and not necessary if the conclusion about transcription is softened. In addition, the effects on maternal RNA in embryos is not easily explained as a transcriptional control and we recommend that this data in embryos be saved for a different paper.

Minor comments and questions?

2. PZ sizes were scored throughout the manuscript without mention of how PZ boundaries were determined. This should be included in the Methods and the PZ boundaries should be marked in the insets in Fig 1A.
3. In figure 1A, the legend explains blue but not gold. Are these pseudocolored? Clarify that main images are 3D models generated in Imaris and that the inset is a confocal image of DAPI staining.
4. Results, first section, first paragraph, line -8. "RNAi for 16 hrs from L4 larval stage". The DTC does not complete migration until the end of L4.
5. Results, first section, second paragraph, line 1. "Expression of *sdn-1*" the words in this sentence suggests that *sdn-1* expression has been assayed in the germline with other methods (e.g. antibodies), just not with transgenes. Was that the intent? Or did you mean that SDN-1 wasn't visible in the germline or somatic gonad?
6. Results, first section, third paragraph, line -4. "We found that unlike wild-type SDN-1, GAG chain-deficient . . . were unable to rescue the PX . . ." Did you check the validity of this negative result by making sure the mutant protein was expressed normally?
7. Results, second section, first paragraph, line 7. "However *glp-1* knockdown did not reduce PZ cell number? Did you check the validity of this negative result by making sure RNAi depleted *glp-1* expression?"
8. Results, second section, first paragraph, line -4. *glp-1(e2141)* is a temperature-sensitive allele but there is no mention of what temperature was used to assay it. Please state the temperature and ensure comparison of controls at the same temperature.
9. Results, second section. Does *trp-2(gk298)* have any reported phenotypes and if yes, could they indirectly affect PZ size?
10. Results, second section, third paragraph. The abstract talks about a "conserved serine in TRPC channels." Is that conserved serine present in the *C. elegans* ortholog?
11. Results, third section, first paragraph, line 1. "We reasoned . . ." This reasoning is not solid. Correlation between RNA and protein levels can be explained by transcriptional or post-transcriptional regulation. Please mention evidence for post-transcriptional control of *glp-1*.
12. Results, third section, first paragraph, line 4. "However transcriptional mechanisms controlling Notch receptor are unknown". This is not correct. There are in fact studies of transcriptional regulation of Notch in other organisms (reviewed in Wu and Bresnick, 2007 for example). In fact, Gopal et al mentions that AP-2 transcription factors interact with the promoter of Notch in the Discussion.
13. Results, third section, first paragraph. "we used RNAi to individually knock down their expression". *aptf-3* and -4 were knocked down together, so individually isn't accurate. Explain why 3 and 4 were knocked down together. In addition, is there redundancy between *aptf-2* and -3 which the authors state have both been detected in the germline (refs 18 and 19)?
14. Results, sixth section, second paragraph, line 6. "We found however that localization of

endogenous APTF-2::GFP expression is unchanged...". We only see the loop region and oocytes in the figure. Is APTF-2 detectable in the progenitor zone? Is localization changed in the progenitor zone?

Figures:

Fig 2 and others: Define CTF in the legend.

Fig 2D Clarify method used to compare protein levels in PZs of different sizes for quantitation (Figure 2D, 3C and E etc.). Are there fewer cells that make the same amount of protein per cell, or does each cell make less protein? The answer to this question is essential to the major conclusions of this paper. Fig 5 F and G and Fig S7. How many times were the EMSA experiments repeated? Are they reproducible?

Fig 5H. For the model figure, it would be helpful to diagram the gonad and which regulators are found in the sheath and the germline to clarify the molecular and anatomical path of signaling from SDN-1 to glp-1 expression.

Reviewer #3 (Remarks to the Author):

The authors find that syndecan-1 expressed in the somatic gonad promotes expression of Notch receptor glp-1 in the germ line by regulating TRP calcium channels controlling Ca²⁺ levels, which affect binding of an AP-2 transcription factor APTF-2 to the glp-1 promoter at an evolutionarily conserved motif.

The work quite comprehensively describes a regulatory pathway upstream of the crucial receptor of the stemness cue in the germ line stem cells. The work is novel, interesting, and important, though the soundness of some of the genetics and quantifications, and their interpretation, is lacking (see below).

I have several major concerns and a few more minor ones about presentation:

1. I have a major concern about the glp-1(e2141) experiment. This is not a glp-1 null allele; it is a temperature sensitive allele that has reduced fertility (and fewer germ cells) at 15°C and 20°C and is sterile at 25°C due to the failure of germ line induction (Priess et al., 1987, Kodoyianni et al., 1992). Upon temp shift from 15 to 25°C of glp-1(ts) mutants that have reached older stages with developed gonads, the mitotic population of the PZ shrinks to nothing over the course of several hours due to the differentiation of all germ cells in the absence of active Notch signaling (Fox and Schedl 2015 used a less severe glp-1(ts) allele for their experiment demonstrating this total shift to meiosis).

In the manuscript under consideration, it is not clear at what temperature these worms were reared. At permissive temperatures the allele is hypomorphic with a relatively mild germ line defect. The presentation reads to me as if the authors consider this to be a glp-1 null allele, which leads to a faulty genetic analysis. Hypomorphs can be deceiving in epistasis tests (see Huang and Sternberg in Wormbook on the importance of null alleles in "Genetic dissection of developmental pathways"). A complete glp-1 loss of function phenotype in the adult germ cells is 0 REC-8(+) PZ cells (see Fox and Schedl 2015), which is a strong enhancement of the sdn-1 phenotype, substantially changing the outcome of the epistasis test as stated: "However, glp-1 knockdown did not reduce PZ cell number of sdn-1(zh20) mutant animals suggesting that sdn-1 and glp-1 act in the same genetic pathway (Figure 2A)".

If these experiments were in fact done with temp shifts, I recommend describing the timing of the temperature shifts for this glp-1(e2141) allele and commenting on the large number of remaining PZ cells at the restrictive temperature which conflicts with prior results. If not, I recommend redoing the experiments with shifts to the restrictive temperature (upon which the cells in the PZ region should differentiate in place), or else removing these experiments from the manuscript because they don't

work in the epistasis test as presented and they undermine the credibility of the work.

It would have been reasonable to investigate whether SDN-1 acted through glp-1 simply based on the known role of glp-1 as a key regulator of the distal germ cells alone, so the failure of this epistasis test is not a fatal flaw. The authors follow up and see that glp-1 transcription and protein abundance measurably respond to sdn-1 loss of function (Figure 2B-D). I'm convinced that sdn-1 acts by regulating glp-1 expression, and simply conclude that sdn-1 contributes to but is not absolutely required for glp-1 expression since the sdn-1 phenotype is less severe than complete glp-1 loss of function.

2. Likewise, the glp-1 RNAi phenotype in Fig. 2A after 16 hour exposure of L4 animals is surprisingly mild. If you use this glp-1 RNAi on your GLP-1::V5 animals, what's the degree of glp-1 knockdown (quantified by loss of detectable GLP-1::V5)? Is the RNAi knockdown just not very penetrant? If not, can the results be trusted? Or is it an issue of cell counting (see next point)?

3. Related to the above points: I'm curious to see all of the quantifications of the germ cell counts—not only how many PZ nuclei but how many TZ and meiotic nuclei were counted by the automatic cell counting program. If the glp-1 RNAi or ts allele caused the germ cells to switch into meiosis, would the cell counting program recognize the abnormally-positioned meiotic germ cells at the distal end of the gonad? I looked at the DB paper by the same group describing this useful automated counting method but found the same ts glp-1(e2141) allele is used again not under temp shift conditions. In Box 1 of that paper it looks like the same size parameters are used to count germ cells in all three regions (PZ, TZ, and meiotic pachytene, but figure 2B suggests different size thresholds?). How does the program distinguish transitions between the three states? Were all samples stained for REC-8? Is there a mathematical description of the "oval" TZ nuclei that finds those and uses them as a landmark? If so what happens when there are no TZ cells?

So my main question about cell counting for the authors is: what does the automated cell counter "see" if you shift L4 glp-1(e2141) worms to 25 degrees for 10+ hours to cause all distal germ cells to enter meiosis? Does it correctly call them all pachytene with 0 PZ and 0 TZ cells? If this test of the automated cell counter fails, then I have serious reservations about the PZ cell number phenotypes reported in the paper.

4. I'm troubled by the differences in the number of wild-type control PZ nuclei in Figure 1B and Figure 2A. There is a difference of ~100 PZ nuclei between these controls, which is substantial—it's analogous to the difference shown in 1B between the wild-type and sdn-1 mutants. 4F control shows less severe but still notable reduction in wild-type.

Do the authors conclude that the difference in bacterial feeding strain (OP50 vs. HT115) alone causes this massive discrepancy? Other differences in the RNAi plates? Developmental staging differences not currently described? Were different portions of the gonad quantified?

Because counting PZ cells is the key phenotypic assessment that the authors use in many experiments, results like this cast doubt on the quality of all of the measurements made. If that large of a difference can arise between controls for unknown reasons, all of a sudden I worry that there are confounding factors at work driving some of the supposedly meaningful differences that are observed. For example, if it's a timing issue that causes these controls to be so different, do the authors confirm that the sdn-1 mutants progress through development with the same timing as the wild-type controls?

More minor presentation concerns:

5. Figure 5D shows nuclear localization of tagged APTF-2 near the bend of the germ line, a region of

the germ line where Notch signaling is not active. It would be better to show nuclear localization for the control and *sdn-1* mutant in the distal germ cells where the GLP-1::V5 reporter is shown, as in Figure 2C.

6. I disagree with the sign of one of the inferred regulatory interactions, which might be my confusion over the diagram, in which case the diagram should be clarified. The summary diagram of Figure 5H shows:

GAG-SDN-1 —| TRP-2 —> APTF-2 —> *glp-1* txn —> PZ proliferation

Shouldn't the sign be TRP-2 —| APTF-2 ?

The epistasis test of the single and double *sdn-1* and *trp-2* mutants and the experiment described here suggests a negative regulatory interaction between Ca²⁺ influx via TRP-2 and the binding of APTF-2: "We found that binding of APTF-2::V5 to the *glp-1* promoter, but not a control protein-DNA interaction, was inhibited by an increase Ca²⁺ levels (Figure 5F-G)."

This diagram should be clarified to show the inhibitory action of TRP-2-mediated Ca²⁺ influx on APTF-2 promoter binding.

7. The quantification of GLP-1::V5(delta promoter) in Figure 3C and 3E. There is substantially more GLP-1::V5 detected in the *glp-1*(delta promoter) than in the *sdn-1*(zh20) mutant in figure 2C. Could this experiment in 3C be repeated side by side with the *sdn-1*(zh20) mutant? Preferably in a blinded fashion? The manuscript only definitively states that the promoter deletion fails to enhance the *sdn-1* defect in PZ cell number not GLP-1::V5 expression, but since expression regulation is the hypothesized mechanism of action I think this needs to be clarified.

8. Rhetorical point: The discussion states, "our study reveals a mechanism where SDN-1 acts as a conduit for somatic-germline communication" but I would argue "conduit" isn't really the right term here. My understanding is that the authors have found that GAG-SDN-1 regulates TRP-2 in the somatic gonad, and TRP-2 regulates Ca²⁺ in the somatic gonad. They infer that this SDN-1-TRP-2 interaction affects Ca²⁺ levels in the germ cells where the Ca²⁺-sensitive binding of APTF-2 to the *glp-1* promoter occurs, possibly through gap junctions. In this model, I'd call the gap junctions the "conduit"; as the work as presented still leaves a question of how the signal is relayed to from the sheath to the germ cells.

Subjective impressions:

I'm quite excited about the findings of this study; there has long been evidence for a pro-proliferative cue from the sheath to the germ cells, and the first site-of-action experiment of this paper identifies SDN-1 regulating such a signal and goes on to characterize most of its mechanism of action, including a previously unidentified regulatory region of the Notch receptor *glp-1*, the key regulator of the stem cell fate in the germ line. I believe these results and am excited for the field to build off of these important findings.

However, the genetic analysis seems to have been done without a strong grounding in (nor sufficient citation of--please look at Fox and Schedl 2015) the copious literature about the role of *glp-1* in the germ line. Furthermore, I'm not convinced that the automated cell counting can detect the range of known germ line phenotypes caused by defects in the *glp-1* pathway. So I'm torn, both believing and feeling excited about the main findings—that *sdn-1* in the gonadal sheath regulates *glp-1* expression in the germ line—but seeing that they were arrived at through unsound genetics that makes me unsure if I can trust the rest of the findings. I happen to have experience with *glp-1*(ts) alleles and scoring germ line defects, but I don't have experience working with HEK or mouse fibroblast cells. If I worry I can't trust the results of the worm genetics that I know quite well, I can't help but be worried about the rest of the findings that I know less about technically but find initially quite convincing.

Comments about statistics and reproducibility:

I would like to see explanations for the discrepancy in the wild type PZ cell count mentioned in point 4 above as they speak to reproducibility, and clarification about the ability of the automated cell counter to distinguish between mitotic and meiotic germ cells in germ lines shifted into meiosis that lack the regionalization of wild-type germ lines mentioned in point 3. While the method for the cell counting is in the DB paper, until I know more about how it calls germ cell state it's hard for me to comment on reproducibility.

I have a statistics question about 3E sdn-1 RNAi on the promoter deletion background (far right column): there is a single high outlier, in fact it's the highest CTF measurement made in this experiment. If this outlier is omitted, what does that do to the determination of non-significance between the control and sdn-1 RNAi groups? Given that RNAi penetrance is often <100%, there are usually some individuals in an RNAi experiment that are just not affected and fall well within the wild-type range. The distribution of points shown in this last column looks to be almost bimodal, with a cluster that closely matches the control and another that is shifted down to a mean of ~ 0.6 . Can the authors comment on this please?

REVIEWER COMMENTS

Reviewer #1 (Remarks to the Author):

Summary:

Gopal and co-authors present convincing and very elegant evidence implicating Syndecan in the transcriptional control of Notch receptor expression to regulate germ cell number in the PZ/mitotic zone of the germline in *C.elegans*. Mutants of Syndecan (2 independent mutations), post-developmental depletion of Syndecan by RNA interference, and rescue of the germline defects with single-copy insertion of Syndecan expressed in the somatic gonad sheath cells, all conclusively establish a role for Syndecan in this context. Importantly, double loss of Syndecan and Notch receptor has similar consequences as the loss of each of the single gene losses, suggesting that Syndecan and the Notch receptor may function in the same pathway. Notch receptor expression level, both mRNA and protein, depends on functional Syndecan.

To define the mechanism regulating germ cell number, the authors successfully identify a novel Notch receptor transcriptional regulatory sequence (TFPA2 motif), and find that the germline expressed transcription factor APTF-2 is required for regulating Notch receptor expression and germ cell number in vivo (using *aptf-2* RNA interference and mutants), as well as competitively binding the TFPA2 motif of the Notch receptor locus in biochemical assays. Deletion of the TFAP2 motif does not enhance the *aptf-2* depletion defects, consistent with a direct regulation, which the authors confirm by EMSA. Further, the authors find that APTF-2 regulation of the Notch receptor requires Syndecan. Finally, the authors show that Ca²⁺ regulates APTF-2 Binding to the *glp-1* promoter in biochemical assays.

Given that in other contexts Syndecan can impact cell behavior by repressing transient receptor potential canonical (TRPC) channel function, resulting in limited cellular Ca²⁺ influx, the authors explore a role for this mode of action of Syndecan (via TRPC) in the control of proliferation in the germline, by studying the sole TRP channel expressed in the gonad of *C.elegans*, TRP2. The authors find that mutation of TRP2 suppresses the reduced levels of GLP-1::V5 expression levels and PZ germ cell number observed in Syndecan mutants, implicating TRP2 in this context, which is suggestive that Syndecan effects could be through Ca²⁺ levels in this biological context as well.

The authors also offer evidence linking the regulatory role of Syndecan, potentially via the gonad-expressed Ca²⁺ channel TRP2, and propose that Syndecan's impact on TRP2 affects Ca²⁺ levels, resulting in changes on transcriptional activity of APTF-2. Since APTF-2 transcriptional activity has been found to be regulated by Ca²⁺ in other contexts, the authors test if regulation of Notch receptor expression (GLP-1::V5) and PZ germ cell number by APTF-2 depends on SDN-1/TRP2. They measure GLP-1::V5 levels in wild-type and Syndecan mutants upon *aptf-2* reduction by RNA interference, which did not further reduce Notch receptor levels compared to the low levels present in Syndecan mutants, in agreement with the possibility that Syndecan acts in the same genetic pathway as *aptf-2* and *glp-1* to control PZ germ cell number.

The authors further elegantly demonstrate that suppression of *sdn-1* germ cell number defect by loss of TRP2 is dependent on the binding of APTF-2 to *glp-1* promoter (as deletion of APTF2 binding site in the *glp-1* promoter prevents suppression of *sdn-1* by TRP2 mutation). This is suggestive that modulation by SDN-1-TRP2 may impact APTF-2 regulation of *glp-1* Notch receptor.

Evaluation:

Overall, this is an excellent and exciting piece of work. The experiments are carried out with great care, using state-of-the-art approaches, powerfully combining in vivo molecular genetics with biochemical and cell biological approaches. The manuscript is well written and the figures are well crafted. The work solidly demonstrates that the proteoglycan Syndecan impacts Notch receptor expression in the germline, and that an AP-2 transcription factor (APTF-2) to the *glp-1* promoter is the mediator of *glp-1* transcriptional regulation. While functional links between Syndecan and Notch signaling have been previously reported, the definition of transcriptional control elements/ pathway for the Notch receptor, implicating the proteoglycan Syndecan, is of great importance to the field of stem cells and Notch. The link between Syndecan and transcription of *glp-1*/Notch receptor is proposed to be via Ca²⁺ levels (as APTF-2 depends on Ca²⁺ for its binding to the promoter of *glp-1*), and would be affected by the somatic TRP calcium channel, be regulated by Syndecan. The demonstration for this part of the pathway is less strong.

Key points of the mechanism remain unaddressed, and the only alternative discussed is that TRP2 and Syndecan are functioning in the same cells, i.e., the somatic gonad. The expression and localization of TRP2 is not investigated (besides citing that it is expressed in the somatic gonad), Ca²⁺ levels in the germline are not directly assayed (in Syndecan or TRP2 mutants, or double), and the localization of Syndecan is not explored (it functions from the somatic gonadal sheath cells as its expression in/from these cells suffices for function, but does Syndecan remain at the somatic cell surface, or is it shed into the germline to directly act on the germ cell nuclei?)

A few experiments (especially points 1-3 below) would strengthen the conclusions and improve the manuscript; other issues could be clarified by text changes.

1) Concerning the mechanism by which Syndecan may impact Notch receptor expression, via TRP2/Ca²⁺. Syndecan function impacting germline is recapitulated by expression in gonadal sheath alone. In the Results summary paragraph, it is claimed that "control of a TRP2 channel by the transmembrane proteoglycan SDN-1 regulates GLP-1 expression", and the assumption of Syndecan acting at the gonad membrane remains (e.g., in first paragraph of Discussion and in Abstract).

It may be that Syndecan indeed remains transmembrane bound at the somatic cell surface, yet there is no investigation of this. An alternative is that Syndecan may be shed from the gonad sheath cells (as it is in numerous contexts). It is conceivable that, once shed, Syndecan could directly act on germ cells or on the syncytial plasma membrane surrounding germ cells, and that the effect of TRP2 may be through regulating the shedding of Syndecan. Syndecan shedding/localization could be investigated by visualizing the N-terminus and the C-terminus of Syndecan with different fluorescent labels.

Related to this, what precise somatic gonad cells express TRP2 and where is it required to function to impact APTF2-mediated Notch receptor transcriptional expression? Can TRP2 mutant defects be rescued by expression under the same driver as used to rescue Syndecan mutants (gonadal sheath = *lim-7* promoter)? i.e., do Syndecan and TRP2 function in the same cells?

Is TRP2 expression affected by Syndecan mutation?

A more basic control here is to ask whether gonad sheath cells integrity is altered in Syndecan mutants? Images show that gonad diameter is smaller and that PZ zone length is reduced (Fig S1). Is somatic gonadal sheath cell morphology (or their processes) altered in Syndecan mutant? Using a fluorescent driver to visualize the gonadal sheath cell in Syndecan mutants would address this important issue.

Syndecan expression in DTC does not rescue, which may be surprising as it can function non-autonomously. A comment would be helpful (this is not so important.)

Reply: We generated a GFP reporter driven by a 5061 bp *sdn-1* promoter. The reporter shows that *sdn-1* is expressed in the gonadal sheath and DTC (Figure 1E-F). We also show that the defect in the *sdn-1* mutant germline is rescued by expressing syndecan in the gonadal sheath (Figure 2H).

Our previous research showed that expression of full length, uncleaved syndecan on the cell surface is required to control TRP channels (PMID: 26391658). Our current data shows the expression of syndecan in the germline using *mex-5* promoter (Figure 2H) did not rescue the defects in the *sdn-1* mutant germline. This suggests that syndecan is not acting directly on germ cells and that SDN-1 shedding is unlikely to control GLP-1 expression. Nonetheless, we obtained a strain expressing SDN-1 with a fluorescent protein tag to visualize the protein. However, the expression levels are so low (as previously reported PMIDs: 30093412, 30416012) in the gonad associated tissues that the visualization of shed syndecan was not feasible.

Reply: We used confocal imaging to examine a fluorescent reporter driven by a *trp-2* promoter and found that it is expressed throughout the gonadal sheath. New images are now included in Figure 2F

Reply: We expressed *trp-2* cDNA in the gonadal sheath using the *lim-7* promoter in the *sdn-1; trp-2* double mutant. We found that *trp-2* expression in the gonadal sheath reduced GLP-1 expression and PZ cell number in the *sdn-1; trp-2* double mutant (Figure 2G-H). Hence, SDN-1 and TRP-2 act in the same tissue to regulate the germline.

Reply: Our previous work showed that syndecans do not affect the expression of TRP channels (PMID: 26391658). In the current manuscript, we analysed the expression pattern of the *trp-2* fluorescent reporter after *sdn-1* silencing. We did not detect changes in *trp-2* localization in the gonadal sheath following *sdn-1* knockdown (Figure 2F).

Reply: We analysed the gonadal sheath morphology by expressing GFP in the sheath of wild-type and *sdn-1* mutant animals. The integrity of gonad sheath remains intact in the absence of *sdn-1* (Figure 1I).

Reply: Regarding the non-rescue of the syndecan phenotype from the DTC. The reason for this is unclear, however the clear rescue of *sdn-1* in the gonadal sheath prompted us to follow this avenue of investigation.

2) About the mechanistic relationship between Syndecan and the Notch receptor: The authors have shown that Notch receptor levels (mRNA and protein) are affected by mutation of Syndecan, and define a transcriptional pathway. However, whether Syndecan mutation affects the activity of the Notch receptor has not been addressed. This seems particularly important since Syndecan is known to impact receptor activity in other contexts.

The authors *could* express *glp-1*/Notch receptor (using a germ line specific driver) to test for suppression of the germ cell defects in Syndecan mutant *sdn-1(zh20)*. Such overexpression of Notch receptor bypassing the need for Syndecan would provide evidence that the Notch receptor acts downstream of Syndecan, and that Syndecan affects the germ cells fate by modulating *glp-1* expression and not its activity.

Reply: We analysed the active Notch receptor by imaging the nuclear localization of active notch intracellular domain (Reviewer Figure 1). We indeed found the NICD localization is reduced in the nuclei, which suggest reduced activity. However, this was not surprising as the total expression of notch receptor is also reduced in the *sdn-1* mutant. Therefore, we focussed on the expression levels of GLP-1, rather than the activity that may be a secondary effect of reduced expression. These data are not included in the revised manuscript.

Reply: Previous reports have shown that the increase in GLP-1 function leads to over proliferation of germ cells, resulting in a germline without distinct zones. Our data shows that syndecan mutants have lower expression of GLP-1, but it is not completely absent. Overexpressing GLP-1 in the mutants may cause a germline tumour (PMID: 9043073). Therefore, it is likely that the expression of GLP-1 will not only suppress the defect but may create new defects. As such, we did not perform this experiment.

3) That mutation of TRP2 suppresses the reduced levels of GLP-1::V5 expression levels and PZ germ cell number observed in Syndecan mutants suggests that Syndecan effects may be through Ca²⁺ levels given its role on Ca²⁺ in other contexts. Additional independent evidence for a role for Ca²⁺ regulation in germ cell development is necessary. To assess Ca²⁺ levels directly, perhaps a similar Ca²⁺ sensors to that used by this group in *C. elegans* in Gopal et al, J. Cell Biol 2015 could strengthen the link with Ca²⁺ per se in this context? Alternatively, optogenetic manipulation of a Ca²⁺ channel may be a way to directly show an impact on Notch receptor transcription?

Related to this, in Fig 5 F, EMSA assays are used to show Ca²⁺ dependency of for APTF2 binding to the *glp-1* promoter. Is this Ca²⁺ effect on APTF-2 binding to the *glp-1* promoter dose-dependent?

Testing whether *inx-8* and *inx-9* indeed allow for germline Ca²⁺ levels regulation by gonadal sheath cells would add context to the mechanism.

Reply: We agree with the reviewer. We exhaustively attempted to express calcium sensors in germline over last six years to assay calcium levels. While we were successful to express other genes in the germline, the worm

silenced the calcium sensors. We tried many advanced genetic tricks to avoid silencing including removal of piRNA sites in the sensor, single copy gene integration by MinMos/MosSCI, adding long introns into sensor gene sequences, different UTRs and the combination of all of these methods. In addition, we injected calcium sensor dyes into the germline, which was unreliable due to high background and uneven distribution. Therefore, we employed EMSA to study the direct impact of calcium on APTF-2 binding to *glp-1* promoter.

To further show the GLP-1 expression is affected by calcium, we silenced an unrelated calcium regulator *stim-1* in *C. elegans*. STIM-1 is a calcium sensor, which activates store operated calcium channels. Silencing *stim-1* increased the expression of GLP-1, likely due to the lower calcium levels resulting from reduced store activated channel activity. This showed that the expression of GLP-1 is dependent on calcium regulators (Reviewer Figure 2). However, we do not deem it necessary to include this in the manuscript.

Reply: We performed a dose dependent assay for APTF-2 binding to *glp-1* promoter. We performed EMSA with calcium ranging from 5 nM to 100 nM. We found that the interaction was disrupted when calcium concentration is ≥ 25 nM (Reviewer figure 3). As there is no gradient but rather a switch when calcium levels increase, we did not include these data in the manuscript.

Reply: We silenced *inx-9* in the germline and analysed GLP-1 expression and mitotic cell number (Reviewer figure 4). We found the mitotic cell number and GLP-1 expression in the *sdn-1* mutant germline were rescued by removing *inx-9*. This suggests that increased calcium in the *sdn-1* mutant germline is likely to be affected by *inx-9* silencing. Although, knockdown of *inx-9* in wild-type also led to a defective germline. Together, it suggests that a balanced function of *inx-9* is required for germ cell development (Reviewer Figure 4). We find the data are not conclusive enough to state that the *innexins* play part in *sdn-1* mediated calcium control, though it remains a possibility as addressed in the discussion.

4) The double loss of Syndecan and Notch receptor have similar consequences as the single gene losses, arguing that Syndecan and the Notch receptor function in the same pathway. Authors should comment on the nature of the Notch receptor allele *glp-1*(e2141) loss-of-function used here (null, partial?), as it is key for data interpretation.

Reply: We removed the *glp-1* mutant data from the manuscript as suggested by reviewer 3. This does not alter the conclusions of the manuscript.

5) That *aptf-2* reduction by RNA interference did not further reduce Notch receptor levels compared to the low levels present in Syndecan mutants, is consistent with Syndecan acting in the same genetic pathway as *aptf-2* and *glp-1* to control PZ germ cell number.

However, testing the triple depletion in this context (*aptf-2* RNAi in *sdn-1*; *glp-1* double mutants, which the authors use earlier in the manuscript) would be important to conclude about the pathway.

Reply: As above, due to the inconclusive nature of the *glp-1* temperature-sensitive mutant, we decided to remove it from our analysis. This doesn't change the main outcomes and conclusion of the manuscript.

6) The model figure depicts only molecules without any cellular context. Reusing the diagram of Fig-1e would help contextualize the genes and molecules implicated.

Reply: Figure 5H is now edited to reflect the reviewer's comments.

7) Fig. 1e should include a representation of the gonadal sheath processes onto the proximal part of the PZ (authors write "gonadal sheath consists of somatic cells that cover the surface of the germline meiotic region and also send processes into the proximal part of the PZ 22-24)

Reply: The figure is now modified according to new literature to show gonadal sheath extension closer towards distal end (PMID: 32692313).

8) Syndecan HS chains appear necessary for Syndecan function on controlling germ cells. It would have been interesting to test if HS chemical modification are also involved in the process. Authors do mention in the Discussion that "Drosophila melanogaster, reducing 3-O sulfation causes Notch-associated neurogenic phenotypes and lower levels of Notch protein".

It could have been tested using well established mutations of *C. elegans* or discuss a potential impact in Discussion.

Reply: We agree that HS modification is an important aspect of proteoglycan signalling. A colleague is currently working on the roles of HS chain modifications in germline development in detail and will submit their data independently. We didn't want to tread on their toes in this manuscript.

9) Related to this, the authors write "Further, SDN-1 requires the attachment of extracellular sugar chains to control germ cell fate, suggesting that extracellular ligands, environmental cues or matrix proteins control this SDN-1/TRP-2 axis." An alternative possibility is that Syndecan devoid of HS chains may not be properly localised/shed?

Reply: We previously examined the role of HS chains in cell surface localization of syndecans (PMID: 20154082). We found that HS chains do not affect the ability of syndecans to localize on the cell surface, and hence we believe that this is unlikely.

10) This sentence is a bit confusing: "We found that unlike in wildtype animals, no reduction in PZ germ cell number was observed in sdn-1(zh20) mutant animals following (Figure 5B)."
Maybe reword to something like "We found that RNAi knockdown of aptf-2, which in wildtype reduces PZ germ cell number, did not further reduce PZ germ cell number in sdn-1 mutant animals"

Reply: The text is edited to reflect reviewer's comments (page 11).

11) Incomplete sentence at page 9 ? "Whereas, sdn-1 RNAi reduces GLP-1::V5 expression and PZ cell number in animals that harbor a wild-type glp-1 promoter (Figure 3E-F)."

Reply: The text is edited to reflect reviewer's comments (page 9).

12) The PZ cell number scale changes between graphs and figures, would be easier to compare results if kept constant across all figures (max= 400)?

Reply: The scales are now uniform in all figures.

Reviewer #2 (Remarks to the Author):

Review of "A Somatic Proteoglycan Controls Notch-Directed Germ Cell Fate"

Overview

Gopal and colleagues report a series of experiments to propose that *C. elegans* syndecans, SDN-1, promotes germline proliferation non-autonomously. They provide evidence that SDN-1 expressed in the somatic gonad regulates a Ca⁺⁺ channel, and that changes in Ca⁺⁺ influence binding of the APTF-2 transcription factor and its transcriptional activation of the *glp-1* gene, which encodes a Notch receptor responsible for maintenance of germline stem cells. The various lines of data provided in their manuscript support each of the steps of this model, and their data is strong for the most part and the manuscript is well written. Some assertions in the manuscript are overly strong and should either be softened or bolstered with additional data. Nonetheless, syndecans regulation of Notch expression is an exciting advance and will be of broad interest.

Major Comments

1. The manuscript would be greatly improved with a clear statement or model figure explaining how SDN-1 in somatic sheath cells affect GLP-1 expression in distal germ cells. The most distal sheath cell processes have marginal contact with distal-most germ cells in the progenitor zone. Where is SDN-1 affecting the calcium change and how is that calcium affecting germ cells in the progenitor zone? Is the idea that calcium diffuses through the syncytium? Also, are there any effects on the more proximal germ cells that have extensive contact with the sheath?

Reply: We have now revised the model figure as requested.

Recent studies have shown that a thin, membranous protrusions of somatic gonad (Sh1 pair) reach to the DTC that control cell divisions and stemness exit within the mitotic region (PMID: 32692313). Therefore, it is conceivable that SDN-1 controls the germ cells in the mitotic region while expressed on the gonadal sheath adjacent to the distal end. Please note comment 7 from reviewer 1.

Ca²⁺ diffuses to the germline through gap junctions in the sheath. To examine the potential role of gap junctions, we performed RNAi against *inx-9* (a gap junction component expressed on the gonadal sheath) and analysed the impact on the germline (Reviewer figure 4). Please refer Point 3C from reviewer 1 for details.

Finally, we did not detect any overt defects in proximal germ cells in *sdn-1* mutant animals.

2. The manuscript reports SDN-1 regulation of *glp-1* expression and states that it occurs at the transcriptional level. For example, the Abstract states: "Hence, SDN-1 signaling promotes GLP-1 transcription and mitotic germ cell fate." Evidence for SDN-1 regulation of *glp-1* expression is compelling but evidence that it occurs the transcriptional level is suggestive rather than compelling. Moreover, the well-established post-transcriptional *glp-1* regulation is not mentioned but should be.

Reply: We have now discussed the role of post-transcriptional control of *glp-1* on pages 13-14. However, our combined evidence supports transcriptional regulation of *glp-1* by APTF-2. Below, we address each point made regarding the role of APTF-2 in controlling *glp-1* transcription.

Below are lines of evidence used to argue for transcriptional control, and the reasons they are not definitive.

(1) Argument: APTF-2 homologs are well established transcription factors and hence imply transcriptional regulation. Caveat: Some canonical transcription factors also bind RNA and regulate RNA activity, probably at the level of stability or translation; most others have not yet been investigated in this light but they cannot be assumed to be limited to transcriptional regulation.

Reply: AP transcription factors have been studied in mammals and other organisms. We have not found any reported post-transcriptional roles for the APTFs. In addition, we have shown that APTF-2 binds to the *glp-1* promoter. Please see point 2 below for details. We have however modified the discussion to address the possibility of post-transcriptional regulation.

(2) Argument: A *glp-1* promoter mutation that deletes an APTF-2 binding site reduces *glp-1* expression. Caveat: The location of the binding site relative to the transcription start site or the 5' end of the 5'UTR should be mentioned to bolster the argument that the binding site lies in the promoter. Is it possible that the binding site resides within the 5'UTR or in the nascent transcript that has not yet been trans-spliced? These may seem like ad hoc considerations, but given the prevalence of post-transcriptional regulation in the *C. elegans* progenitor zone, they are worth considering.

Reply: We examined the TSS mapping data of the *C. elegans* genome (PMID: 23636945). We found that the *glp-1* TSS and APTF-2 binding site are 208bp and 221bp upstream of the start codon, respectively. This proximity solidifies a role for APTF-2 in the transcriptional control of *glp-1* and we have added this information in the manuscript on page 9.

(3) Argument: APTF-2 binds the consensus site in the *glp-1* promoter in vitro. Caveat: In vitro binding of a DNA binding protein to its consensus site is not surprising, but it does not mean that APTF-2 binds that site in the *C. elegans* germline? Can the authors find evidence for in vivo binding in available Chip-Seq data? Although such data would not address APTF-2 action in the progenitor zone, it would bolster the argument.

Reply: We exhaustively attempted ChIP-PCR to detect the interaction between APTF-2 and the *glp-1* promoter *in vivo*. We found that after lysing worms, APTF-2::GFP degraded within minutes. We were unable to prevent this by using either protease inhibitors or altering buffer conditions. In addition, the level of expression of APTF-2 in the

germline is extremely low. Therefore, we used a well-established *in vitro* method (EMSA) to show that APTF-2 can interact with conserved promoter region in the *glp-1* promoter. Importantly, this interaction was removed when the APTF binding site was mutated.

(4) Argument: Using qPCR to test RNA extracted from *sdn-1* mutants and wildtype worms, *glp-1* mRNA levels were lower in the mutants. Caveat: The vast majority of *glp-1* mRNA is in oocytes, where it accumulates as maternal RNA for deposition in embryos; much less is present in the progenitor zone. This qPCR test is therefore unlikely to measure expression changes in the progenitor zone.

Reply: As the reviewer alluded to, in the qPCR experiments RNA was extracted from dissected germlines that contained ~3 oocytes. Therefore, as the reviewer mentions, we measured *glp-1* mRNA from the entire germline including oocytes. This suggests that *sdn-1* regulates *glp-1* in germline and/or oocytes. To show the *glp-1* mRNA is reduced specifically in the germline of *sdn-1* mutant animals, we have now performed smFISH and quantified the level of *glp-1* mRNA the germline excluding oocytes (Figure S4). We found that *glp-1* mRNA is indeed reduced in the non-oocyte regions of *sdn-1* mutant germline.

(5) Argument: Expression was tested with single molecule FISH to visualize *glp-1* RNAs and with immunostaining to visualize GLP-1 protein cells in wildtype and *glp-1* “promoter” mutant animals and embryos. *glp-1* mRNA was lower in mutant embryos but could not be tested in the progenitor zone due to background; GLP-1 protein was lower in the mutant progenitor zone. The lowered GLP-1 protein does indeed show an effect on expression but not on transcription. Caveat 1: The rigorous way to assay transcription in the progenitor zone is to visualize active transcription sites (ATS) in progenitor zone nuclei and to determine numbers of ATS and number of nascent transcripts at ATS. Effects on the abundance of cytoplasmic transcripts may reflect post-transcriptional rather than transcriptional regulation. ATS analyses in the progenitor zone are well established with smFISH and a publicly available MATLAB code. It was not clear from the Methods section if smFISH was performed in dissected gonads or whole worms. The standard method uses dissected gonads. Caveat 2: *glp-1* mRNA in embryos is maternal and its differential expression is known to be subject to post-transcriptional regulators. Although differences were reported in the abundance of those maternal RNAs, those differences may not reflect transcriptional control. Recommendation: The authors could either do additional experiments and measure transcription directly in the progenitor zone (ATS analyses) or soften their conclusion with respect to transcriptional control. The latter strategy would require revisions throughout the text but no new experiments. Their current data clearly show that SDN-1 and the APTF-2 binding site influence *glp-1* expression and this is very interesting. The simplest explanation is transcriptional control but it is not a slam dunk. We recommend that the smFISH experiments be repeated in dissected gonads to report cytoplasmic mRNAs in the progenitor zone without background, which would be a useful addition though not as definitive as an ATS analysis. The ATS analysis is more challenging and not necessary if the conclusion about transcription is softened. In addition, the effects on maternal RNA in embryos is not easily explained as a transcriptional control and we recommend that this data in embryos be saved for a different paper.

Reply: Please see new smFISH quantification above (Figure S4). Since *glp-1* mRNA detected highly in the embryos, we chose to analyze the expression in the embryo. Now we have removed the embryo data and added germline specific data (Figure S4). We would like to reiterate that APTF-2 regulation of *glp-1* expression may not be limited to PZ. Secondly, the identification of the APTF binding site at the TSS of *glp-1* strongly supports a transcriptional effect.

Reply: We thank the reviewer for the insightful comments here. To reflect these important points, we detailed the possibility of post-transcriptional modification in the discussion and soften the language regarding a purely transcriptional role where appropriate.

Minor comments and questions?

2. PZ sizes were scored throughout the manuscript without mention of how PZ boundaries were determined. This should be included in the Methods and the PZ boundaries should be marked in the insets in Fig 1A.

Reply: We should have made this clearer. The PZ/TZ boundary was determined manually when two crescent shaped cells were identified. This detailed methodological information has now been incorporated in full into the supplementary material. Page 5.

3. In figure 1A, the legend explains blue but not gold. Are these pseudocolored? Clarify that main images are 3D models generated in Imaris and that the inset is a confocal image of DAPI staining.

Reply: Changes are made in the figure and text on page 22.

4. Results, first section, first paragraph, line -8. “RNAi for 16 hrs from L4 larval stage”. The DTC does not complete migration until the end of L4.

Reply: We thank the reviewer for pointing it out. The text is now edited to reflect reviewers’ comments on page 5. Previous reports showed that *sdn-1* mutant animal shows a variable DTC migration path, however, the DTCs do reach the correct final position (PMID: 19631636). To confirm these data however, we have now performed experiments to silence *sdn-1* in the knockdown timeframe used for germline analysis. This experiment revealed no defects in the DTC migration following *sdn-1* knockdown (Figure S1).

5. Results, first section, second paragraph, line 1. “Expression of *sdn-1*” the words in this sentence suggests that *sdn-1*

expression has been assayed in the germline with other methods (e.g. antibodies), just not with transgenes. Was that the intent? Or did you mean that SDN-1 wasn't visible in the germline or somatic gonad?

Reply: We actually meant to say that *sdn-1* expression has not been reported in the germline or associated tissues previously by either transgenes or antibodies. We now showed by using a fluorescent reporter that syndecan is expressed in the gonadal sheath and distal tip cells (Figure 1E-F).

6. Results, first section, third paragraph, line -4. "We found that unlike wild-type SDN-1, GAG chain-deficient . . . were unable to rescue the PX . . ." Did you check the validity of this negative result by making sure the mutant protein was expressed normally?

Reply: Please refer to point 9 from reviewer 1.

7. Results, second section, first paragraph, line 7. "However *glp-1* knockdown did not reduce PZ cell number? Did you check the validity of this negative result by making sure RNAi depleted *glp-1* expression?"

Reply: Yes. We now include a graph to show the *glp-1* knockdown. Figure S3.

8. Results, second section, first paragraph, line -4. *glp-1(e2141)* is a temperature-sensitive allele but there is no mention of what temperature was used to assay it. Please state the temperature and ensure comparison of controls at the same temperature.

Reply: We have now omitted *glp-1* mutant data from the manuscript.

9. Results, second section. Does *trp-2(gk298)* have any reported phenotypes and if yes, could they indirectly affect PZ size?

Reply: The *trp-2(gk298)* allele has no reported germline phenotypes. Minor variations in body bend and locomotion were observed in a high-throughput behavioral screen (PMID 23852451). We have now rescued the *trp-2(gk298)* mutant germline phenotype by expressing *trp-2* cDNA in the gonadal sheath – revealing its focus-of-action as the same as SDN-1.

10. Results, second section, third paragraph. The abstract talks about a "conserved serine in TRPC channels." Is that conserved serine present in the *C. elegans* ortholog?

Reply: Yes - this serine residue is conserved

11. Results, third section, first paragraph, line 1. "We reasoned . . ." This reasoning is not solid. Correlation between RNA and protein levels can be explained by transcriptional or post-transcriptional regulation. Please mention evidence for post-transcriptional control of *glp-1*.

Reply: We have now mentioned post-transcriptional control of *glp-1* on pages 13-14.

12. Results, third section, first paragraph, line 4. "However transcriptional mechanisms controlling Notch receptor are unknown". This is not correct. There are in fact studies of transcriptional regulation of Notch in other organisms (reviewed in Wu and Bresnick, 2007 for example). In fact, Gopal et al mentions that AP-2 transcription factors interact with the promoter of Notch in the Discussion.

Reply: Yes. We meant to write "in *C. elegans*" so have now amended this sentence

13. Results, third section, first paragraph. "we used RNAi to individually knock down their expression". *aptf-3* and *-4* were knocked down together, so individually isn't accurate. Explain why 3 and 4 were knocked down together. In addition, is there redundancy between *aptf-2* and *-3* which the authors state have both been detected in the germline (refs 18 and 19)?

Reply: There is significant sequence overlap between *aptf-3* and *-4*. APTF-3 was previously reported to be expressed in the germline, but not APTF-4 (PMID: 30093412). The RNAi clone used (*aptf-3*) in this experiment can knockdown both *aptf-3* and *-4* due this sequence similarity. We have now performed qPCR that confirms the RNAi clone we used indeed reduces expression of both *aptf-3* and *-4* (Reviewer figure 5). In addition, we generated new RNAi clones that independently target *aptf-3* and *-4* to confirm they do not play a role in germline function (Reviewer figure 5). Finally, *aptf-2* and *-3* were simultaneously silenced to exclude possibility of the redundancy (Reviewer figure 5).

14. Results, sixth section, second paragraph, line 6. "We found however that localization of endogenous APTF-2::GFP expression is unchanged...". We only see the loop region and oocytes in the figure. Is APTF-2 detectable in the progenitor zone? Is localization changed in the progenitor zone?

Reply: The expression level of APTF-2::GFP is very low in the PZ region, which precludes reliable quantification. In addition, APTF-2 is unstable after germline dissection, making it impossible to stain with an antibody. Therefore, we chose a region in which we could accurately observe APTF-2 expression in live animals.

Figures:

Fig 2 and others: Define CTF in the legend.

Reply: Now included full definition of CTF in the Methods section.

Fig 2D Clarify method used to compare protein levels in PZs of different sizes for quantitation (Figure 2D, 3C and E etc.). Are there fewer cells that make the same amount of protein per cell, or does each cell make less protein? The answer to this question is essential to the major conclusions of this paper.

Reply: The CTCF measurement for GLP-1 expression has now been validated with a secondary measurement. CTCF was re-measured for 5 individual cells positioned at a same distance from the distal end of the germline. The results are consistent with the previous version of the manuscript and each cell expresses reduced GLP-1 protein. (Reviewer figure 6).

Fig 5 F and G and Fig S7. How many times were the EMSA experiments repeated? Are they reproducible?

Reply: Experiments were repeated >3 times and EMSA is validated using by biotin labelled and unlabelled probe. The experiment is now repeated with a calcium gradient (reviewer figure 3).

Fig 5H. For the model figure, it would be helpful to diagram the gonad and which regulators are found in the sheath and the germline to clarify the molecular and anatomical path of signaling from SDN-1 to glp-1 expression.

Reply: Figure is now modified to reflect reviewers' comments.

Reviewer #3 (Remarks to the Author):

The authors find that syndecan-1 expressed in the somatic gonad promotes expression of Notch receptor glp-1 in the germ line by regulating TRP calcium channels controlling Ca²⁺ levels, which affect binding of an AP-2 transcription factor APTF-2 to the glp-1 promoter at an evolutionarily conserved motif.

The work quite comprehensively describes a regulatory pathway upstream of the crucial receptor of the stemness cue in the germ line stem cells. The work is novel, interesting, and important, though the soundness of some of the genetics and quantifications, and their interpretation, is lacking (see below).

I have several major concerns and a few more minor ones about presentation:

1. I have a major concern about the glp-1(e2141) experiment. This is not a glp-1 null allele; it is a temperature sensitive allele that has reduced fertility (and fewer germ cells) at 15°C and 20°C and is sterile at 25°C due to the failure of germ line induction (Priess et al., 1987, Kodoyianni et al., 1992). Upon temp shift from 15 to 25°C of glp-1(ts) mutants that have reached older stages with developed gonads, the mitotic population of the PZ shrinks to nothing over the course of several hours due to the differentiation of all germ cells in the absence of active Notch signaling (Fox and Schedl 2015 used a less severe glp-1(ts) allele for their experiment demonstrating this total shift to meiosis).

In the manuscript under consideration, it is not clear at what temperature these worms were reared. At permissive temperatures the allele is hypomorphic with a relatively mild germ line defect. The presentation reads to me as if the authors consider this to be a glp-1 null allele, which leads to a faulty genetic analysis. Hypomorphs can be deceiving in epistasis tests (see Huang and Sternberg in Wormbook on the importance of null alleles in "Genetic dissection of developmental pathways"). A complete glp-1 loss of function phenotype in the adult germ cells is 0 REC-8(+) PZ cells (see Fox and Schedl 2015), which is a strong enhancement of the sdn-1 phenotype, substantially changing the outcome of the epistasis test as stated: "However, glp-1 knockdown did not reduce PZ cell number of sdn-1(zh20) mutant animals suggesting that sdn-1 and glp-1 act in the same genetic pathway (Figure 2A)".

If these experiments were in fact done with temp shifts, I recommend describing the timing of the temperature shifts for this glp-1(e2141) allele and commenting on the large number of remaining PZ cells at the restrictive temperature which conflicts with prior results. If not, I recommend redoing the experiments with shifts to the restrictive temperature (upon which the cells in the PZ region should differentiate in place), or else removing these experiments from the manuscript because they don't work in the epistasis test as presented and they undermine the credibility of the work.

It would have been reasonable to investigate whether SDN-1 acted through glp-1 simply based on the known role of glp-1 as a key regulator of the distal germ cells alone, so the failure of this epistasis test is not a fatal flaw. The authors follow up and see that glp-1 transcription and protein abundance measurably respond to sdn-1 loss of function (Figure 2B-D). I'm convinced that sdn-1 acts by regulating glp-1 expression, and simply conclude that sdn-1 contributes to but is not absolutely required for glp-1 expression since the sdn-1 phenotype is less severe than complete glp-1 loss of function.

Reply: We totally agree with the reviewer here. We also conclude that sdn-1/aptf-2 contribute to regulation of glp-1 but are not absolutely required as the phenotype of sdn-1 and aptf-2 are less severe than a complete loss of glp-1 function. As suggested, we have removed our attempt to perform epistasis analysis from the manuscript using the temperature-sensitive glp-1 allele.

Reply: We thank the reviewer for pointing out the discrepancy in the figure. It was an error on the Y-axis of figure 2A, which prompted concerns from the reviewer. The figure is now corrected.

2. Likewise, the glp-1 RNAi phenotype in Fig. 2A after 16 hour exposure of L4 animals is surprisingly mild. If you use this glp-1 RNAi on your GLP-1::V5 animals, what's the degree of glp-1 knockdown (quantified by loss of detectable GLP-1::V5)? Is the RNAi knockdown just not very penetrant? If not, can the results be trusted? Or is it an issue of cell counting (see next point)?

Reply: We included a new graph to show the validity of knockdown (Figure S3). Note that this is a short-term RNAi, which is unlikely to abolish GLP-1 expression. Within in 16h of RNAi treatment, we achieved approximately 60% knockdown of GLP-1 (new Figure S3).

3. Related to the above points: I'm curious to see all of the quantifications of the germ cell counts—not only how many PZ nuclei but how many TZ and meiotic nuclei were counted by the automatic cell counting program. If the glp-1 RNAi or ts allele caused the germ cells to switch into meiosis, would the cell counting program recognize the abnormally-positioned meiotic germ cells at the distal end of the gonad? I looked at the DB paper by the same group describing this useful automated counting method but found the same ts glp-1(e2141) allele is used again not under temp shift conditions. In Box 1 of that paper it looks like the same size parameters are used to count germ cells in all three regions (PZ, TZ, and meiotic pachytene, but figure 2B suggests different size thresholds?). How does the program distinguish transitions between the three states? Were all samples stained for REC-8? Is there a mathematical description of the "oval" TZ nuclei that finds those and uses them as a landmark? If so what happens when there are no TZ cells?

So my main question about cell counting for the authors is: what does the automated cell counter "see" if you shift L4 glp-1(e2141) worms to 25 degrees for 10+ hours to cause all distal germ cells to enter meiosis? Does it correctly call them all pachytene with 0 PZ and 0 TZ cells? If this test of the automated cell counter fails, then I have serious reservations about the PZ cell number phenotypes reported in the paper.

Reply: The software does not define each region (PZ, TZ, meiotic pachytene). This was defined manually, based on the nuclear shape. Please see reply to reviewer 2. The scoring parameters for the analysis are

defined based on the microscopy technique, objective and camera zoom - detailed now in full in the Supplementary materials page 5.

4. I'm troubled by the differences in the number of wild-type control PZ nuclei in Figure 1B and Figure 2A. There is a difference of ~100 PZ nuclei between these controls, which is substantial—it's analogous to the difference shown in 1B between the wild-type and *sdn-1* mutants. 4F control shows less severe but still notable reduction in wild-type.

Do the authors conclude that the difference in bacterial feeding strain (OP50 vs. HT115) alone causes this massive discrepancy? Other differences in the RNAi plates? Developmental staging differences not currently described? Were different portions of the gonad quantified?

Because counting PZ cells is the key phenotypic assessment that the authors use in many experiments, results like this cast doubt on the quality of all of the measurements made. If that large of a difference can arise between controls for unknown reasons, all of a sudden I worry that there are confounding factors at work driving some of the supposedly meaningful differences that are observed. For example, if it's a timing issue that causes these controls to be so different, do the authors confirm that the *sdn-1* mutants progress through development with the same timing as the wild-type controls?

Reply: We thank the reviewer for pointing out the discrepancy in the figure. It was an error on the Y-axis of figure 2A, which prompted concerns from the reviewer. The figure is now corrected.

More minor presentation concerns:

5. Figure 5D shows nuclear localization of tagged APTF-2 near the bend of the germ line, a region of the germ line where Notch signaling is not active. It would be better to show nuclear localization for the control and *sdn-1* mutant in the distal germ cells where the GLP-1::V5 reporter is shown, as in Figure 2C.

Reply: Previous reports showed expression of *aptf-2* mRNA in the distal end which increases towards the proximal end (PMID: 30093412). We found it very challenging to image APTF-2::GFP at the distal end, likely due to very low endogenous expression. Nevertheless, we show that the nuclear localization of APTF-2::GFP is unchanged in *sdn-1* mutant animals. Please refer reviewer 2 minor comment 14 for details.

6. I disagree with the sign of one of the inferred regulatory interactions, which might be my confusion over the diagram, in which case the diagram should be clarified. The summary diagram of Figure 5H shows:

GAG-SDN-1 —| TRP-2 —> APTF-2 —> *glp-1* txn —> PZ proliferation

Shouldn't the sign be TRP-2 —| APTF-2 ?

The epistasis test of the single and double *sdn-1* and *trp-2* mutants and the experiment described here suggests a negative regulatory interaction between Ca²⁺ influx via TRP-2 and the binding of APTF-2: “We found that binding of APTF-2::V5 to the *glp-1* promoter, but not a control protein-DNA interaction, was inhibited by an increase Ca²⁺ levels (Figure 5F-G).”

This diagram should be clarified to show the inhibitory action of TRP-2-mediated Ca²⁺ influx on APTF-2 promoter binding.

Reply: We have modified the diagram to show the locations of each protein in the germline and their regulatory hierarchy.

7. The quantification of GLP-1::V5(delta promoter) in Figure 3C and 3E. There is substantially more GLP-1::V5 detected in the *glp-1*(delta promoter) than in the *sdn-1*(zh20) mutant in figure 2C. Could this experiment in 3C be repeated side by side with the *sdn-1*(zh20) mutant? Preferably in a blinded fashion? The manuscript only definitively states that the promoter deletion fails to enhance the *sdn-1* defect in PZ cell number not GLP-1::V5 expression, but since expression regulation is the hypothesized mechanism of action I think this needs to be clarified.

Reply: We did not observe a difference between figure 3C and 3E GLP-1::V5(delta promoter). Each experiment was performed with its own controls and the CTCF normalized against either wild-type or RNAi controls. This nullifies the variations in the intensity levels between each replicate that may arise from the subtle changes in the microscopes. This is elaborated in method section now. Supplementary materials page 5.

Changes in secondary antibody (new vs old) and life of the microscope's UV source are the possible reasons for the different appearance in the image 2C and 3. We have now performed additional staining side-by-side as suggested by reviewer to compare wild-type, *sdn-1*(zh20), *sdn-1*(zh20); *trp-2*(gk298) and *glp-1-prom delta* (Reviewers Figure 7). This confirms our findings.

8. Rhetorical point: The discussion states, “our study reveals a mechanism where SDN-1 acts as a conduit for somatic-germline communication” but I would argue “conduit” isn't really the right term here. My understanding is that the authors have found that GAG-SDN-1 regulates TRP-2 in the somatic gonad, and TRP-2 regulates Ca²⁺ in the somatic gonad. They infer that this SDN-1-TRP-2 interaction affects Ca²⁺ levels in the germ cells where the Ca²⁺-sensitive binding of APTF-2 to the *glp-1* promoter occurs, possibly through gap junctions. In this model, I'd call the gap junctions the “conduit”; as the work as presented still leaves a question of how the signal is relayed to from the sheath to the germ cells.

Reply: The text is edited on page 15 and 28 to reflect reviewers' comments. Please refer reply to reviewers 1 and 2 for new gap junction data.

Subjective impressions:

I'm quite excited about the findings of this study; there has long been evidence for a pro-proliferative cue from the sheath to the germ cells, and the first site-of-action experiment of this paper identifies SDN-1 regulating such a signal and goes on to characterize most of its mechanism of action, including a previously unidentified regulatory region of the Notch receptor glp-1, the key regulator of the stem cell fate in the germ line. I believe these results and am excited for the field to build off of these important findings.

However, the genetic analysis seems to have been done without a strong grounding in (nor sufficient citation of--please look at Fox and Schedl 2015) the copious literature about the role of glp-1 in the germ line. Furthermore, I'm not convinced that the automated cell counting can detect the range of known germ line phenotypes caused by defects in the glp-1 pathway. So I'm torn, both believing and feeling excited about the main findings—that sdn-1 in the gonadal sheath regulates glp-1 expression in the germ line—but seeing that they were arrived at through unsound genetics that makes me unsure if I can trust the rest of the findings. I happen to have experience with glp-1(ts) alleles and scoring germ line defects, but I don't have experience working with HEK or mouse fibroblast cells. If I worry I can't trust the results of the worm genetics that I know quite well, I can't help but be worried about the rest of the findings that I know less about technically but find initially quite convincing.

Comments about statistics and reproducibility:

I would like to see explanations for the discrepancy in the wild type PZ cell count mentioned in point 4 above as they speak to reproducibility, and clarification about the ability of the automated cell counter to distinguish between mitotic and meiotic germ cells in germ lines shifted into meiosis that lack the regionalization of wild-type germ lines mentioned in point 3. While the method for the cell counting is in the DB paper, until I know more about how it calls germ cell state it's hard for me to comment on reproducibility.

I have a statistics question about 3E sdn-1 RNAi on the promoter deletion background (far right column): there is a single high outlier, in fact it's the highest CTF measurement made in this experiment. If this outlier is omitted, what does that do to the determination of non-significance between the control and sdn-1 RNAi groups? Given that RNAi penetrance is often <100%, there are usually some individuals in an RNAi experiment that are just not affected and fall well within the wild-type range. The distribution of points shown in this last column looks to be almost bimodal, with a cluster that closely matches the control and another that is shifted down to a mean of ~0.6. Can the authors comment on this please?

Reply: The concern regarding genetic analysis is addressed. The temperature sensitive glp-1 data is now removed as suggested by the reviewer (Reviewer 3- Point 1 and 2). Please see above for automated counting explanation (Reviewer 3, point 3 and 4).

Reply: Figure 3E control and sdn-1 (bars 3 and 4) have one outlier value each. Removing both or one of the outliers do not change the result interpretation. We included a figure by removing the outliers for reviewer's reference (Reviewer figure 8).

REVIEWER COMMENTS

Reviewer #2 (Remarks to the Author):

Conveyed to the editors:
Criticisms have been appropriately addressed.

Reviewer #3 (Remarks to the Author):

The authors have addressed my major concern by removing analyses including the *glp-1(ts)* allele. Other clarifying experiments—especially quantification *glp-1* RNAi knockdown efficacy using the *glp-1::V5* allele—have also improved this interesting and important study. My outstanding scientific questions and then more minor text suggestions follow.

As for the differences I raised among controls in different germ cell count experiments, the authors state that the Y-axis in Figure 2A was incorrect, and the figures are much improved by harmonizing the scale of Y-axes across all germ cell count figure panels.

However, Figure 2A doesn't appear to have only an adjustment of the axis. For comparison, the graph in Figure 3F that originally had a max Y-value of 300 has been rescaled to have a max Y-value of 400—the point clouds of the measurements for each treatment are clearly just stretched/compressed along the Y axis, as expected. Comparing Figure 2A in the prior and current submission, the point clouds for each treatment have different shapes (unfortunately I don't have access to an excel sheet from the original submission, so I can't compare the data points directly). I encourage the authors to double check which data are plotted. If these are different data than shown in the original submission, some justification seems necessary.

I'm confounded by the new smFISH experiment shown in Figure S4. As you have shown, GLP-1 protein is abundant in the distal end of the germ line and indeed is known to function there. Why are you unable to detect transcript in that region? Why do you use that region as a control?

On a related note, in replying to Reviewer 2, the authors state: "We would like to reiterate that APTF-2 regulation of *glp-1* expression may not be limited to PZ."

I again suggest that the manuscript be specific and well-referenced with respect to its argument about how and where *glp-1* acts in the germ line in the model being presented. Is there any detectable *glp-1* transcription in the bend region where APTF-2 measurements are made? How would APTF-2 regulation of *glp-1* in this region—where no germ cell proliferation occurs—affect the size of the PZ?

Reviewer 2 raises interesting questions about transcriptional vs. post-transcriptional control of *glp-1*. I have less of a problem with the inference that the control is transcriptional, but I also think a transcriptional reporter could be informative here. Because the germline has so much post-transcriptional regulation and the GLP-1 protein has a PEST sequence, a *glp-1* wild type or mutant promoter driving GFP with no PEST and a neutral 3' UTR (that won't be targeted by GLD-1) might be a useful (and simple) reagent to test the effect on transcription of some of the manipulations done in the paper.

Below I include some final suggestions mostly pertaining to the written clarity of the arguments being presented.

- Thank you for clarifying how your cell counting was done. I recommend the following text changes

for precision (*new text* Xto removeX):

163 based on nuclei shape. The *proximal* end of the proliferation zone was defined as immediately
164 XpriorX *distal* to detection of the *distal-most row of germ cells with* two crescent shaped
nuclei, marking the start of the transition zone.

Line 166 misspelling: mitotic

Line 66 "secondary messenger" should be "second messenger"

- The below sentence (while it's an interesting and important hypothesis!) is misplaced in the results section; this is a discussion point that is not tested.

157 Our

158 finding that SDN-1 functions in gonadal sheath cells to control germ cell behavior
159 suggests that SDN-1 may interpret extra-germline signals (e.g. environmental and
160 matrix cues) at the germline-soma interface.

- I encourage the following be explained more thoroughly with respect to the duration of RNAi treatment and partial knockdown of glp-1.

176 We *achieved partial knock down of* glp-1 16
177 hours from the L4 stage using RNAi (complete knock down causes the differentiation of all PZ cells
FOX AND SCHEDL) and analyzed young adult hermaphrodites (Figure
178 S3). As shown previously, decreased glp-1 expression in wild-type animals reduces PZ
179 cell number (Figure 2A)8. However, glp-1 *RNAi treatment* did not *further* reduce PZ cell
number of

- The added text on lines 187-193 refers to the glp-1::V5 allele, but comes right before the glp-1::V5 allele is introduced, so rearranging would be helpful.

- Please clarify this statement. Should it end with "...in the nervous system"? Or "...do not affect total mRNA expression levels of TRP channel genes"?

220 Our previous study showed that
221 syndecans do not affect the expression of TRP channels³.

- This is lab slang
227 these readouts

- In this statement, "positively" is unclear/incorrect because of the inhibitory relationships between SDN-1 and TRP-2 and between TRP-2 and glp-1.

231 SDN-1/TRP-2 regulatory axis XpositivelyX regulates glp-1 expression

- Significance brackets in 2G and 2H—could these be staggered to show which comparison matches which significance asterisks?

- I recommend the following change for precision, since this is an immunostaining experiment:

279 We found that aptf-2 RNAi

280 reduced GLP-1::V5 XexpressionX *protein detected* in the distal end of the germline (Figure 4B-C).

- Because “these data” don’t address the role of calcium (that’s in the next section), perhaps rephrase:

326 Together, these data posit that modulation of Ca²⁺ levels

327 by the SDN-1/TRP-2 axis controls APTF-2-dependent regulation of PZ germ cell

328 number by GLP-1, likely through modulation of Ca²⁺ levels.

Reviewer #4 (Remarks to the Author):

I read through the revised manuscript and the responses to the original reviewer's comments. I think that the authors have made a good-faith effort and have addressed the majority of the reviewers comments. The only question I have is the author's choice not to include the stim-1 and to a lesser extent, the EMSA data in the supplemental material. I agree that these are probably not essential to the main point of the manuscript, but does provide supporting contextual information that might be appreciated by some readers.

REVIEWER COMMENTS

Reviewer #4 (Remarks to the Author):

Conveyed to the editors:
Criticisms have been appropriately addressed.

Reviewer #3 (Remarks to the Author):

The authors have addressed my major concern by removing analyses including the *glp-1(ts)* allele. Other clarifying experiments—especially quantification *glp-1* RNAi knockdown efficacy using the *glp-1::V5* allele—have also improved this interesting and important study. My outstanding scientific questions and then more minor text suggestions follow.

As for the differences I raised among controls in different germ cell count experiments, the authors state that the Y-axis in Figure 2A was incorrect, and the figures are much improved by harmonizing the scale of Y-axes across all germ cell count figure panels. However, Figure 2A doesn't appear to have only an adjustment of the axis. For comparison, the graph in Figure 3F that originally had a max Y-value of 300 has been rescaled to have a max Y-value of 400—the point clouds of the measurements for each treatment are clearly just stretched/compressed along the Y axis, as expected. Comparing Figure 2A in the prior and current submission, the point clouds for each treatment have different shapes (unfortunately I don't have access to an excel sheet from the original submission, so I can't compare the data points directly). I encourage the authors to double check which data are plotted. If these are different data than shown in the original submission, some justification seems necessary.

As mentioned in the previous revision, original Figure 2A had an error on the Y-axis which has been resolved. When re-creating the graph in prism with a Y-value of 400 the software reconfigured the positioning of the dot points. However, this does not change the actual data - only its presentation. We have now also uploaded all raw data as required by the journal in our "Source Data" excel file.

I'm confounded by the new smFISH experiment shown in Figure S4. As you have shown, GLP-1 protein is abundant in the distal end of the germ line and indeed is known to function there. Why are you unable to detect transcript in that region? Why do you use that region as a control? On a related note, in replying to Reviewer 2, the authors state: "We would like to reiterate that APTF-2 regulation of *glp-1* expression may not be limited to PZ." I again suggest that the manuscript be specific and well-referenced with respect to its argument about how and where *glp-1* acts in the germ line in the model being presented. Is there any detectable *glp-1* transcription in the bend region where APTF-2 measurements are made? How would APTF-2 regulation of *glp-1* in this region—where no germ cell proliferation occurs—affect the size of the PZ?

We totally understand this point raised by the reviewer. Previously published data showed that the spatial expression of *glp-1* mRNA is abundant at the proximal germline (which is repressed post-transcriptionally), and lowly detected at the distal end (PMID: 30416012; PMID: 30093412) - see image below from (PMID: 30416012) - left is distal, right is proximal.

Our smFISH experiments confirmed this expression profile for *glp-1* mRNA in Fig S4 i.e. high detection of *glp-1* mRNA proximally. Therefore, we selected the more proximal regions to measure *glp-1* by smFISH as the distal expression was not able to be reliably quantified. In order to normalize the fluctuations in the fluorescence levels, we chose the least bright region of the germline as the background. The same logic was followed to show APTF-2 expression. Our data and previous publications have shown that APTF-2 has a similar expression pattern as *glp-1* mRNA (PMID: 30416012; PMID: 30093412), that is very low expression at the distal end and high expression at the proximal end. These references are included in the manuscript. Our measurements of *glp-1* mRNA by smFISH do not suggest in any way that *glp-1* functions in the proximal region to control the size of the PZ. The function for GLP-1 is well-described in the distal region. However, due to technical challenges and the very low *glp-1* mRNA expression in the distal end, we were forced to measure in the proximal region. We have now made this section clearer in the manuscript.

Reviewer 2 raises interesting questions about transcriptional vs. post-transcriptional control of *glp-1*. I have less of a problem with the inference that the control is transcriptional, but I also think a transcriptional reporter could be informative here. Because the germline has so much post-transcriptional regulation and the GLP-1 protein has a PEST sequence, a *glp-1* wild type or mutant promoter driving GFP with no PEST and a neutral 3' UTR (that won't be targeted by GLD-1) might be a useful (and simple) reagent to test the effect on transcription of some of the manipulations done in the paper.

We have multiple lines of evidence to show that the identified mechanism in this paper regulates *glp-1* transcription (as confirmed by this reviewer and Reviewer 2). This includes mutation of the *glp-1* promoter at the transcriptional start site (TSS), which reduces *glp-1* expression. We also showed that the APTF-2 transcription factor interacts with this promoter region - an interaction that is dependent on this transcription factor binding site. Further, to our knowledge, AP transcription factors have no known function besides transcriptional regulation. Together, these data show that *glp-1* is regulated transcriptionally. We therefore believe that we sufficiently confirmed the transcriptional regulation by multiple experiments.

The reviewer suggests the generation of new genetic reagents to re-confirm the transcriptional control of *glp-1*. We believe that these are unnecessary. In addition, the generation of the suggested genetic tools are not straightforward, or guaranteed to work, and will not provide additional insight into the mechanism we have revealed in this manuscript.

Below I include some final suggestions mostly pertaining to the written clarity of the arguments being presented.

• Thank you for clarifying how your cell counting was done. I recommend the following text changes for precision (*new text* Xto removeX):

163 based on nuclei shape. The *proximal* end of the proliferation zone was defined as immediately
164 XpriorX *distal* to detection of the *distal-most row of germ cells with* two crescent shaped nuclei, marking the start of the transition zone.

Corrected.

Line 166 misspelling: mitotic

Corrected.

Line 66 “secondary messenger” should be “second messenger”

Corrected.

• The below sentence (while it’s an interesting and important hypothesis!) is misplaced in the results section; this is a discussion point that is not tested.

157 Our

158 finding that SDN-1 functions in gonadal sheath cells to control germ cell behavior
159 suggests that SDN-1 may interpret extra-germline signals (e.g. environmental and
160 matrix cues) at the germline-soma interface.

Agreed. We removed this sentence as we already have a similar sentence at the end of the Discussion.

• I encourage the following be explained more thoroughly with respect to the duration of RNAi treatment and partial knockdown of *glp-1*.

176 We *achieved partial knock down of* *glp-1* 16

177 hours from the L4 stage using RNAi (complete knock down causes the differentiation of all PZ cells FOX AND SCHEDL) and analyzed young adult hermaphrodites (Figure

178 S3). As shown previously, decreased *glp-1* expression in wild-type animals reduces PZ

179 cell number (Figure 2A)8. However, *glp-1* *RNAi treatment* did not *further* reduce PZ cell number of

Corrected.

• The added text on lines 187-193 refers to the *glp-1::V5* allele, but comes right before the *glp-1::V5* allele is introduced, so rearranging would be helpful.

Corrected.

• Please clarify this statement. Should it end with “...in the nervous system”? Or “...do not affect total mRNA expression levels of TRP channel genes”?

220 Our previous study showed that

221 syndecans do not affect the expression of TRP channels3.

Corrected.

• This is lab slang

227 these readouts

Corrected.

• In this statement, “positively” is unclear/incorrect because of the inhibitory relationships between SDN-1 and TRP-2 and between TRP-2 and *glp-1*. 231 SDN-1/TRP-2 regulatory axis XpositivelyX regulates *glp-1* expression

Corrected.

• Significance brackets in 2G and 2H—could these be staggered to show which comparison matches which significance asterisks?

Corrected.

• I recommend the following change for precision, since this is an immunostaining experiment:

279 We found that *aptf-2* RNAi

280 reduced *GLP-1::V5* XexpressionX *protein detected* in the distal end of the germline (Figure 4B-C).

Corrected.

• Because “these data” don’t address the role of calcium (that’s in the next section), perhaps rephrase:
326 Together, these data posit that modulation of Ca²⁺ levels
327 by the SDN-1/TRP-2 axis controls APTF-2-dependent regulation of PZ germ cell
328 number by GLP-1, likely through modulation of Ca²⁺ levels.
Corrected.

Reviewer #4 (Remarks to the Author):

I read through the revised manuscript and the responses to the original reviewer's comments. I think that the authors have made a good-faith effort and have addressed the majority of the reviewers comments. The only question I have is the author's choice not to include the stim-1 and to a lesser extent, the EMSA data in the supplemental material. I agree that these are probably not essential to the main point of the manuscript, but does provide supporting contextual information that might be appreciated by some readers.

We do not think that adding the stim-1 data (which were part of a reviewers figure) would provide any additional mechanistic insight to our study as it is under-developed. Therefore, we do not wish to include these data to the manuscript.

In response to requests for further elaboration of suboptimal conditions:

The original dissection and staining of the germline for Figure 2A generated sub-optimal images and low-quality DAPI staining. As such, the edges of the nuclei were not sharp, and some nuclei had weak staining. This created difficulties for Imaris software to recognize all the nuclei within the PZ region, eventually leading to a smaller nuclei count than expected. Therefore, we repeated this experiment in review which is now presented in the final version of the manuscript.

REVIEWERS' COMMENTS

Reviewer #3 (Remarks to the Author):

The authors have addressed all my concerns (but don't forget to replace "REFs" on lines 191 and 197!) Remaining questions about the relationships between regional glp-1 transcription, protein abundance, and signaling activity serve to highlight the importance of the work reported in this manuscript.